# SynerMedGen: Synergizing Medical Multimodal Understanding with Generation via Task Alignment

Weiren Zhao [1]    Yi Dong [1]    Cheng Chen [1]

## Abstract

Unifying multimodal understanding and generation is a compelling frontier that is beginning to emerge in the medical field. However, the limited existing unified medical models typically treat understanding and generation as disjoint objectives, lacking a meaningful functional synergy. In this work, we identify and address a critical question in unified medical modeling: what form of "understanding" truly benefits generation. We present SynerMedGen, a unified framework built on the proposed principle of generation-aligned understanding, which synergizes understanding objectives with generation tasks via task alignment. SynerMedGen introduces three generation-aligned understanding tasks and a two-stage training strategy that transfers generation-beneficial representations learned during understanding training to medical image synthesis. Remarkably, even with understanding training alone, our SynerMedGen achieves strong zero-shot performance across 22 medical image synthesis tasks and demonstrates robust generalization. When combined with generation training, SynerMedGen consistently outperforms state-of-the-art specialized medical image synthesis models as well as recent unified medical models. We also release SynerMed, a large-scale dataset of 1M paired synthesis samples and 2M understanding instances for studying understanding–generation synergy. Our project can be accessed at https://github.com/piooip/SynerMedGen.

## 1. Introduction

Recent years have witnessed rapid and parallel advances in multimodal understanding and visual generation, driven respectively by large vision-language models (Brown et al., 2020; Hu et al., 2022; Liu et al., 2023; Li et al., 2024) and diffusion-based generative models (Peebles & Xie, 2023; Zhang et al., 2023; Rombach et al., 2022). More recently, this progress has begun to converge in multimodal large language models (MLLMs) that unify understanding and generation within a single framework, enabling models to both interpret complex visual semantics and synthesize visual content (Team, 2024; Xie et al., 2025; Wu et al., 2025a;b; Lu et al., 2024). Despite this momentum, such unified paradigms remain largely underexplored in the medical domain, where they are especially compelling as clinical imaging workflows require interpreting and transforming over heterogeneous modalities such as CT, multi-sequence MRI, PET to support diagnosis, treatment planning, and monitoring (Li et al., 2023b; Tu et al., 2024).

Very recent efforts such as HealthGPT (Lin et al., 2025) and UniMedVL (Ning et al., 2025) have begun to explore unified medical MLLMs that couple medical understanding with image generation. HealthGPT supports both capabilities via separate task-specific adapters, while UniMedVL follows a progressive learning curriculum for understanding and generation. Despite this progress, a core question remains unresolved: *what kind of "understanding" is actually beneficial for medical image generation?* In exisiting frameworks, the understanding branch is typically trained with proxy objectives that are only weakly related to the generation tasks. As illustrated in Figure 1, understanding tasks that emphasize global semantics can be insufficient for medical image generation tasks such as cross-modality synthesis, which requires retaining fine-grained anatomical/pathological content, respecting target-modality constraints, and maintaining pixel-level correspondence. This yields *task misalignment* that a model may become better at answering recognition-style questions, yet still fail to provide generation-beneficial representations (Fan et al., 2025; Huang et al., 2025).

To address this limitation, we propose a simple but general principle: *align understanding with generation by deriving understanding tasks from the generation dataset itself*. We refer to this as a generation-aligned understanding, because the understanding tasks are co-defined with the generation objectives. In medical image generation, cross-modality

---

[1]The University of Hong Kong, Hong Kong, China. Correspondence to: Cheng Chen <cchen@eee.hku.hk>.

*Proceedings of the 43rd International Conference on Machine Learning*, Seoul, South Korea. PMLR 306, 2026. Copyright 2026 by the author(s).

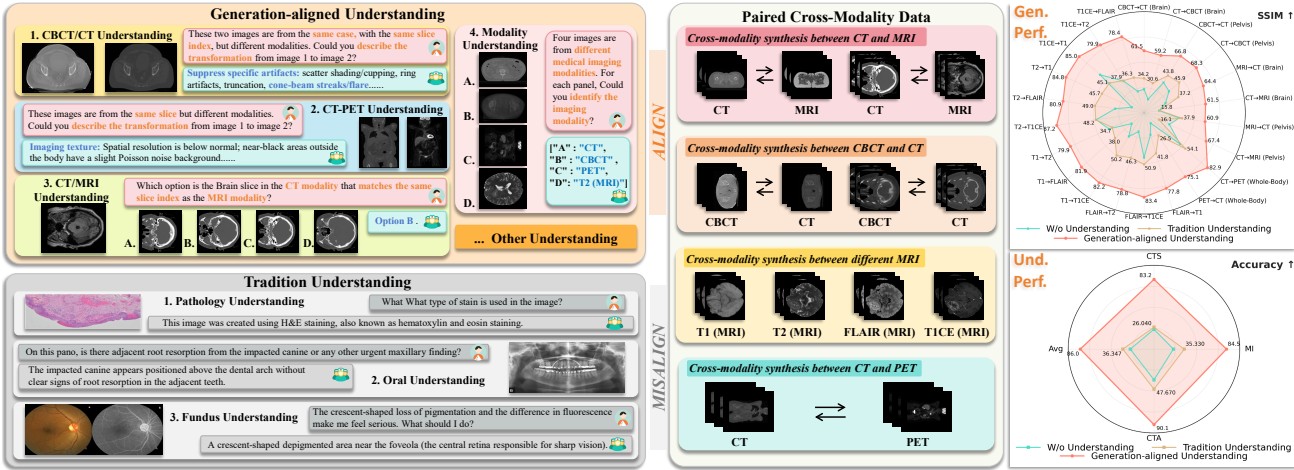

*Figure 1.* Overview of the comparison of the proposed generation-aligned understanding supervision and tradition understanding supervision. The right panel shows a comparison of SSIM values on the synthetic tasks under different understanding settings, as well as a comparison of accuracies on generation-aligned understanding tasks.

image synthesis represents one of the most prevalent and important generation tasks, yet remains highly challenging, as it requires preserving patient-specific anatomy and pathology while modifying only modality-dependent appearance (Reaungamornrat et al., 2022; Kang et al., 2023). Given its representativeness and difficulty, we focus on cross-modality image synthesis as a testbed for validating our design principle of medical understanding and generation model. For cross-modality medical image synthesis, the understanding tasks should facilitate (i) retain invariant clinical content to enable faithful synthesis, (ii) represent modality as an explicit controllable factor rather than an entangled cue, and (iii) encode the direction and constraints of the synthesis so that generation follows the intended modality change rather than an ambiguous appearance shift (Kim et al., 2024; Cohen et al., 2018; Zhang et al., 2018). These criteria are employed to guide the design of our generation-aligned understanding tasks.

Following these design principles, we propose SynerMedGen, a unified framework which synergize medical multimodal understanding with generation via generation-aligned understanding task design. Our SynerMedGen constructs a minimal set of synthesis aligned vision-language tasks directly from paired synthesis data: (1) Conditional Target Selection enforces one-to-one paired correspondence under an explicit target-modality request; (2) Modality Identification makes modality an explicit, learnable control factor; and (3) Transformation Instruction Alignment grounds paired visual differences in text to specify what should change (appearance/contrast) and what must remain invariant (structures/lesions), disambiguating synthesis direction. We employ a two-stage training strategy. First, the model learns these aligned tasks to obtain a generation-beneficial representation. Second, we optimize latent-space conditional synthesis on the same paired data, effectively transferring

the learned multimodal priors into the generation branch.

Our main contributions are highlighted as follows:

- We identify a fundamental misalignment between understanding and generation in existing unified medical MLLMs, and propose SynerMedGen, that explicitly aligns understanding task design with generation objectives to synergize understanding with generation.

- We design three generation-aligned understanding tasks, that effectively learn representations beneficial for medical image synthesis.

- We conduct extensive evaluations of SynerMedGen against state-of-the-arts across 22 tasks, as well as on zero-shot performance, and generalization to unseen datasets, demonstrating strong improvements.

- We release SynerMed, a large-scale dataset containing 1M paired samples and 2M generation-aligned understanding questions.

## 2. Related Work

### 2.1. Medical Multimodal Large Language Models

Early medical MLLMs typically couple a medical vision encoder with a general-purpose LLM through a projection module, enabling strong performance on comprehension-oriented tasks such as VQA and report generation (Li et al., 2023b; Thawakar et al., 2024; Zhang et al., 2024a; Moor et al., 2023; Zhang et al., 2024b). However, most of these systems are not designed to perform medical image synthesis or modality translation within the same model, and generation is commonly delegated to separate diffusion or synthesis pipelines (Li et al., 2023b; Zhang et al., 2025; Meng et al., 2024; Dorjsembe et al., 2024). A parallel line of work

improves medical MLLMs via data and instruction engineering, e.g., scaling medical vision–language pairs, cleaning noisy alignments, and instruction tuning (Lin et al., 2023; Chen et al., 2024). More recently, efforts such as HealthGPT explore unified training and lightweight adaptation to bridge understanding and generation (Lin et al., 2025). Nevertheless, a central challenge remains: the understanding supervision is often not aligned with the information needed for medical images synthesis (e.g., slice-level correspondence, anatomy preservation, and modality-conditioned appearance changes). Our work addresses this limitation by deriving generation-aligned understanding tasks directly from paired synthesis data, so that the learned representation is explicitly optimized to support the generation task.

## 2.2. Unified Multimodal Understanding and Generation

Unified multimodal models in general domains broadly follow three paradigms: (i) autoregressive or diffusion-style generation over discrete image tokens with joint text understanding (Shi et al., 2025; Wang et al., 2025), (ii) dual-pathway designs that separate representations for understanding and generation (Ge et al., 2024; Wu et al., 2025a; Jiao et al., 2025; Tang et al., 2025), and (iii) hybrid/connector-based frameworks that couple language modeling with diffusion or flow-based image synthesis (Xie et al., 2025; Zhou et al., 2025; Ma et al., 2025b; Deng et al., 2025). Recent work further improves tokenization and representation alignment to better balance semantic reasoning and visual fidelity (Qu et al., 2025; Ma et al., 2025a). Despite this progress, directly transferring these paradigms to medical image synthesis remains challenging: clinical synthesis requires stricter control (e.g., anatomy preservation, and slice-level correspondence), which is not reliably enforced by generic multimodal supervision (Koetzier et al., 2024; Thummerer et al., 2023; Shin et al., 2025). Motivated by this gap, we propose SynerMedGen, which derives generation-aligned understanding tasks directly from paired synthesis data and transfers the learned representation into latent-space generation via a two-stage training schedule.

## 3. Method

We first introduce the unified multimodal understanding and generation framework. We then describe the proposed method *SynerMedGen*, which synergizes medical multimodal understanding with generation via generation-aligned understanding task design and transferring the learned representation through a two-stage pipeline. An overview of the method is shown in Figure 2.

### 3.1. Overview of Unified Framework

We build on the Bagel unified architecture Deng et al. (2025), but our method is a general design principle and is ag-nostic to the architecture. Given an input image $x$, an understanding-oriented encoder $E_{\text{ViT}}$ outputs semantic tokens $\mathbf{z}_{\text{ViT}} = E_{\text{ViT}}(x)$, while a generation-oriented encoder $E_{\text{VAE}}$ maps $x$ into VAE latent tokens $\mathbf{z}_{\text{VAE}} = E_{\text{VAE}}(x)$. Two projection layers $f_{\text{ViT}}$ and $f_{\text{VAE}}$ align these tokens to a shared Mixture-of-Transformer-experts (MoT) hidden space. The MoT contains two decoder-style experts with shared infrastructure: an understanding expert for VLM prompted learning over interleaved text and ViT tokens $[\mathbf{x}_{\text{text}}, f_{\text{ViT}}(\mathbf{z}_{\text{ViT}})]$, and a generation expert for conditional synthesis over VAE latents $[f_{\text{VAE}}(\mathbf{z}_{\text{VAE}})]$. For synthesis, the generation expert attends to conditioning evidence (text and/or image tokens) via cross-attention, and a VAE decoder $D_{\text{VAE}}$ reconstructs pixel outputs from the generated latents.

Although a unified framework supports both understanding and generation, it remains unclear what forms of understanding can ensure that shared representations are truly beneficial to the generation process. We therefore propose a generation-aligned understanding to guide the understanding branch learns representations useful for generation.

### 3.2. Generation-Aligned Understanding

Existing unified medical MLLM frameworks often train understanding and generation with separate datasets and objectives, which can lead to *supervision misalignment*. For example, semantic understanding signals learned from recognition-style tasks (e.g., lesion-centric VQA in pathology) may not transfer to cross-modality radiological synthesis, where the model must preserve fine-grained, slice-level patient content. For medical image synthesis, the shared representation should satisfy three requirements: (i) maintain one-to-one correspondence between paired source/target slices, (ii) represent the target modality as an explicit controllable factor, and (iii) capture the direction and constraints of the mapping. Based on this, see Figure 2, we construct three generation-aligned tasks from each pair of samples.

**MoT interface for SynerMedGen tasks.** We formulate all generation-aligned supervision as *prompted generation* on the understanding pathway. Given an image (or an aligned image pair) and a text prompt, we encode the image(s) with $E_{\text{ViT}}$ and feed the interleaved sequence $[\mathbf{x}_{\text{text}}, f_{\text{ViT}}(\mathbf{z}_{\text{ViT}})]$ into the understanding expert. Each task is designed so that the model answers by generating a short string $\mathbf{y}^*$. We train with masked next-token prediction and compute cross-entropy only on the answer tokens:

$$\mathcal{L}_{\text{NTP}}(\mathbf{y}^*) = -\sum_{i=1}^{|\mathbf{y}^*|} \log p_\theta(y_i^* \mid \mathbf{y}_{<i}^*, \mathbf{x}_{\text{text}}, \mathbf{z}_{\text{ViT}}). \quad (1)$$

**Conditional Target Selection (CTS).** CTS trains slice-level correspondence under an explicit target-modality request. Given a source slice $x_{\text{src}}$ and a target modality $m_{\text{tgt}}$, the model must select the paired target slice $x^+ = x_{\text{tgt}}$ from

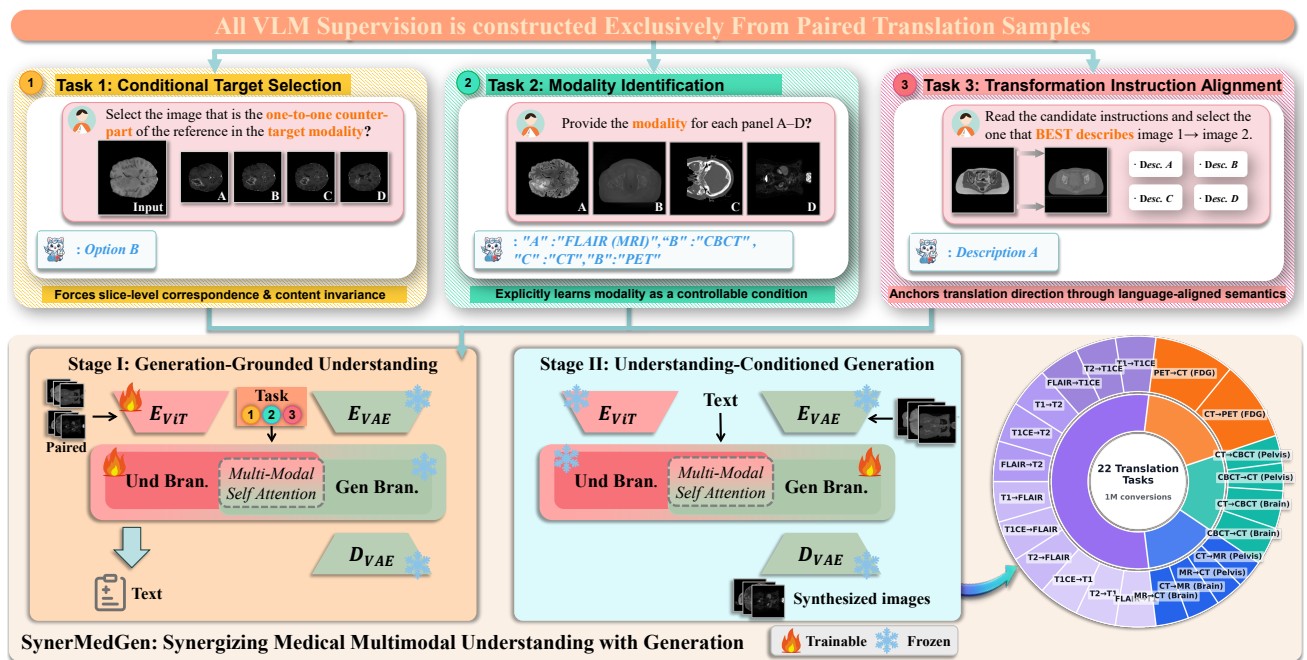

*Figure 2.* **SynerMedGen overview.** From **1M** paired samples, we construct **2M** generation-aligned understanding instances for three tasks: Conditional Target Selection (CTS), Modality Identification (MI), and Transformation Instruction Alignment (TIA). Stage I (GAU) learns a synthesis-sufficient representation; Stage II (UCG) performs flow matching in VAE latent space.

$N$ candidates. To make the task depend on fine-grained alignment (rather than coarse semantics), we sample hard negatives from nearby slices in the same target volume: if the paired target is at index $k$, we draw negatives from $\{x_{\text{tgt}}(k + \delta)\}$ with $\delta \in \{\pm 1, \pm 2, \ldots, \pm K\}$. We format CTS as a multiple-choice prompt and train the model to generate the correct option letter:

$$\mathcal{L}_{\text{CTS}} = \mathcal{L}_{\text{NTP}}(\mathbf{y}^*_{\text{CTS}}), \qquad (2)$$

where $\mathbf{y}^*_{\text{CTS}}$ is the ground-truth option token.

**Modality Identification (MI).** While CTS enforces correspondence, controllable synthesis additionally requires modality to be an explicit factor. MI asks the model to predict the modality of an input image (or each panel), covering CT, CBCT, PET, and MRI (optionally with MRI sequences). We include both easy cases and confusable pairs (e.g., CT vs. CBCT or closely related MR contrasts) to reduce reliance on superficial shortcuts. MI is also implemented as prompted generation, where the model outputs a modality label:

$$\mathcal{L}_{\text{MI}} = \mathcal{L}_{\text{NTP}}(\mathbf{y}^*_{\text{MI}}), \qquad (3)$$

where $\mathbf{y}^*_{\text{MI}}$ is the ground-truth modality label.

**Transformation Instruction Alignment (TIA).** Even with correspondence and modality awareness, synthesis can remain ambiguous without understanding the mapping *direction* and the *constraints* of a route (what should change vs. what should remain invariant). TIA aligns an observed paired change with a short route-level description. For each synthesis route (an ordered modality pair), we build a pool of

concise descriptions and construct a multiple-choice prompt for an aligned pair $(x_1, x_2)$ by sampling one positive description $e^+$ from the ground-truth route pool and $R-1$ distractors from other routes, including common confusions such as swapped direction or mismatched modality pairs. The model is trained to generate the correct option letter:

$$\mathcal{L}_{\text{TIA}} = \mathcal{L}_{\text{NTP}}(\mathbf{y}^*_{\text{TIA}}), \qquad (4)$$

where $\mathbf{y}^*_{\text{TIA}}$ denotes the correct option token.

**Joint objective.** We combine the three tasks into a single understanding-stage loss:

$$\mathcal{L}_{\text{under}} = \mathcal{L}_{\text{CTS}} + \mathcal{L}_{\text{MI}} + \mathcal{L}_{\text{TIA}}, \qquad (5)$$

Together, CTS, MI, and TIA encourage the understanding pathway to preserve slice-level correspondence, represent modality as an explicit control signal, and capture constraints on changes in the synthesis direction—properties directly required for medical image synthesis.

### 3.3. Two-stage Optimization Schedule

SynerMedGen follows a two-stage schedule that explicitly transfers objective-relevant understanding into synthesis. Stage I learns a synthesis-sufficient representation on the understanding pathway using the generation-aligned tasks (CTS/MI/TIA). Stage II then trains latent-space conditional synthesis while conditioning on the stage I representation.

**Stage I: Generation-Aligned Understanding (GAU).** From paired synthesis data, we construct prompted understanding instances for CTS/MI/TIA and optimize the joint

*Table 1.* Quantitative comparison of synthesis methods on SynthRAD2023 and AutoPET with SSIM (PSNR, MAE in Appendix B.1).

| Method | Brain | | Pelvis | | Brain | | Pelvis | | Whole-Body | |
|---|---|---|---|---|---|---|---|---|---|---|
| | CBCT→CT | CT→CBCT | CBCT→CT | CT→CBCT | MRI→CT | CT→MRI | MRI→CT | CT→MRI | CT→PET | PET→CT |
| Pix2Pix (Isola et al., 2017) | 66.17 | 61.34 | 63.55 | 54.34 | 74.33 | 66.88 | 67.11 | 56.32 | 72.21 | 70.89 |
| CycleGAN (Zhu et al., 2017) | 53.32 | 50.91 | 55.87 | 49.44 | 52.65 | 51.26 | 48.31 | 56.09 | 65.98 | 62.03 |
| BBDM (Li et al., 2023a) | 71.09 | 69.20 | 60.49 | 64.19 | 68.99 | 63.61 | 64.37 | 50.86 | 67.68 | 58.15 |
| ResViT (Dalmaz et al., 2022) | 85.00 | 71.03 | 84.00 | 76.32 | 86.39 | 74.26 | 78.77 | 70.31 | 87.07 | 84.50 |
| SynDiff (Özbey et al., 2023) | 85.47 | 70.37 | 83.21 | 78.29 | 87.19 | 73.31 | 78.98 | 72.38 | 88.12 | 86.21 |
| RCD (Li et al., 2025) | 85.97 | 70.92 | 86.22 | 81.34 | 86.12 | 75.56 | 77.09 | 71.37 | 88.90 | 87.34 |
| HealthGPT (Lin et al., 2025) | 57.37 | 43.23 | 46.89 | 54.31 | 84.29 | 75.53 | 78.96 | 71.33 | 66.54 | 58.43 |
| UniMedVL (Ning et al., 2025) | 51.48 | 29.93 | 43.94 | 52.32 | 54.11 | 71.37 | 52.26 | 68.84 | 74.12 | 47.27 |
| SynerMedGen | **87.15** | **73.08** | **87.14** | **83.91** | **88.87** | **77.09** | **81.98** | **74.22** | **91.10** | **88.22** |

*Table 2.* Quantitative comparison of image synthesis methods on BraTS with SSIM (PSNR, MAE in Appendix B.1).

| Method | T1→T2 | T2→T1 | T1→T1C. | T1C.→T1 | T1→FL. | FL.→T1 | T2→T1C. | T1C.→T2 | T2→FL. | FL.→T2 | T1C.→FL. | FL.→T1C. |
|---|---|---|---|---|---|---|---|---|---|---|---|---|
| Pix2Pix (Isola et al., 2017) | 62.10 | 59.34 | 63.17 | 53.04 | 60.79 | 67.78 | 61.36 | 57.93 | 56.24 | 66.88 | 60.04 | 64.20 |
| CycleGAN (Zhu et al., 2017) | 53.31 | 57.19 | 55.34 | 46.76 | 46.02 | 55.78 | 50.35 | 53.79 | 51.30 | 58.43 | 49.29 | 55.80 |
| BBDM (Li et al., 2023a) | 56.41 | 56.77 | 70.08 | 58.60 | 54.07 | 59.27 | 67.07 | 60.38 | 59.64 | 62.23 | 57.01 | 66.37 |
| ResViT (Dalmaz et al., 2022) | 85.78 | 86.94 | 84.84 | 81.39 | 80.69 | 86.52 | 86.44 | 87.64 | 84.47 | 85.21 | 85.00 | 84.15 |
| SynDiff (Özbey et al., 2023) | 84.25 | 88.31 | 86.32 | 82.90 | 81.34 | 87.73 | 88.15 | 87.42 | 87.11 | 85.87 | 86.43 | 88.03 |
| RCD (Li et al., 2025) | 86.19 | 88.01 | 85.34 | 80.70 | 82.91 | 88.45 | 87.33 | 88.19 | 86.79 | 87.57 | 86.01 | 86.26 |
| HealthGPT (Lin et al., 2025) | 70.32 | 60.13 | 63.56 | 63.47 | 69.05 | 60.09 | 67.13 | 73.19 | 66.07 | 68.89 | 64.37 | 66.36 |
| UniMedVL (Ning et al., 2025) | 78.88 | 77.26 | 74.51 | 78.89 | 77.84 | 73.91 | 73.44 | 81.89 | 74.18 | 75.92 | 78.80 | 74.26 |
| SynerMedGen | 87.14 | 90.58 | 89.64 | 85.65 | 85.58 | 90.62 | 92.45 | 90.41 | 87.85 | 89.92 | 88.39 | 90.08 |

loss $\mathcal{L}_{\text{under}}$ in Eq. (5). The model is trained to generate short answers (option letters or modality labels) with next-token prediction using the understanding pathway (including $E_{\text{ViT}}$, $f_{\text{ViT}}$, and the understanding expert). This stage encourages the shared representation to preserve slice-level correspondence, disentangle patient content from modality, and encode mapping direction/constraints.

**Stage II: Understanding-Conditioned Generation (UCG).** We initialize the unified model from stage I and train conditional synthesis via flow matching in VAE latent space. For each paired target image $x_{\text{tgt}}$, we compute the clean latent $z_0 = E_{\text{VAE}}(x_{\text{tgt}})$, sample noise $z_1 \sim \mathcal{N}(0, I)$ and time $t \sim \mathcal{U}(0, 1]$, and form $z_t = (1 - t)z_0 + tz_1$. The generation expert predicts the velocity field conditioned on source evidence:

$$\mathcal{L}_{\text{gen}} = \mathbb{E}_{t, z_1}\left[\|v_\theta(z_t, t, c) - (z_1 - z_0)\|_2^2\right]. \quad (6)$$

Where $v_\theta$ is the prediction network parameterized by the generation expert, The conditioning $c$ is constructed from the source slice $x_{\text{src}}$ and text request (Appendix D provides examples of the text prompts).

## 4. Experiments

### 4.1. Data Structures

**SynerMed.** We build SynerMed (Figure 2), a unified paired dataset for medical image synthesis by integrating multiple public medical synthesis resources, including BraTS (Baid et al., 2021), SynthRAD2023 (Thummerer et al., 2023), and AutoPET (Gatidis et al., 2022). The collection spans multiple anatomical regions, including the brain, pelvis, and abdomen, and various imaging modalities, including CT, MRI, PET, and CBCT. Overall, it contains **1M** paired synthesis samples covering diverse cross-modality and cross-organ synthesis directions. To assess generalization, we additionally evaluate our models on MyoPS (Li et al., 2023c) and SynthRAD2025 (Thummerer et al., 2025) datasets. MyoPS provides paired cardiac images in three modalities (bSSFP, LGE, and T2). SynthRAD2025 includes multi-modal data for the head-and-neck and abdomen.

**Synthesis-aligned understanding tasks.** From the same paired corpus, we construct generation-aligned vision-language questions for stage I training. Specifically, we design three tasks directly from paired synthesis samples, that are CTS for slice-level correspondence under a target-modality request, MI for modality recognition for explicit modality control, and TIA for direction/constraint grounding through text. This yields **2M** training instances for multimodal understanding, designed to provide balanced coverage across organs and modalities and to emphasize fine-grained, translation-critical cues. Examples of understanding questions are provided in Appendix E.

### 4.2. Comparison with State-of-the-arts

**Evaluation protocol.** We evaluate synthesis quality using **SSIM** and **MAE**, with additional **PSNR** results reported in the appendix. For volumetric datasets, we perform slice-wise inference, stack the generated slices back into full 3D volumes according to the original slice order, and compute

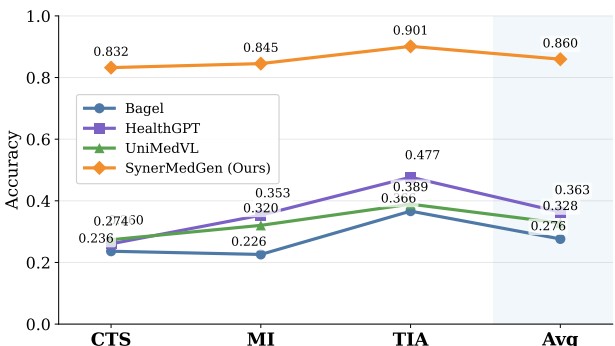

*Figure 3.* Visual question answering accuracy on the three generation-aligned understanding tasks (CTS, MI, TIA).

all quantitative metrics at the 3D volume level for all methods. This protocol ensures that slice-based inference is evaluated consistently under a volume-level setting.

**Comparison on image synthesis.** We conduct extensive comparisons with *(i)* the most recent unified medical multimodal frameworks, including HealthGPT (Lin et al., 2025) and UniMedVL (Ning et al., 2025), which integrate medical visual understanding and generation within a single model, and *(ii)* SOTA specialized medical image synthesis approaches, including general image synthesis approaches Pix2Pix (Isola et al., 2017), CycleGAN (Zhu et al., 2017), and BBDM (Li et al., 2023a), as well as medical-specific synthesis models ResViT (Dalmaz et al., 2022), SynDiff (Özbey et al., 2023), and RCD (Li et al., 2025). We make every effort to ensure a fair and consistent comparison. Since HealthGPT and UniMedVL have not yet released their implementation code, we evaluate these methods using their publicly available checkpoints. Notably, the Brain MRI/CT and Pelvis MRI/CT datasets used in our experiments are also employed in HealthGPT, and the BraTS dataset is also used in UniMedVL. For specialized medical image synthesis methods, we retrain each model using their released code on the same datasets and tasks as those used in our method.

As shown in Table 1 and Table 2, our SynerMedGen consistently outperforms all comparison methods across all cross-modality image synthesis tasks. When compared with the unified multimodal frameworks HealthGPT and UniMedVL, our SynerMedGen achieves significantly better performance no matter on datasets included in HealthGPT and UniMedVL's training, that are Brain and Pelvis MRI/CT for HealthGPT, and BraTS for UniMedVL, or other datasets, validating the effectiveness of our generation-aligned understanding strategy in improving cross-modality image synthesis. Compared with strong specialized medical image synthesis methods ResViT, SynDiff, and RCD, our SynerMedGen demonstrate consistent improvements, highlighting the advantage of our synergistic understanding and generation framework for universal cross-modality image synthesis. For instance, on BraTS MRI T2 → T1CE, our SynerMed-

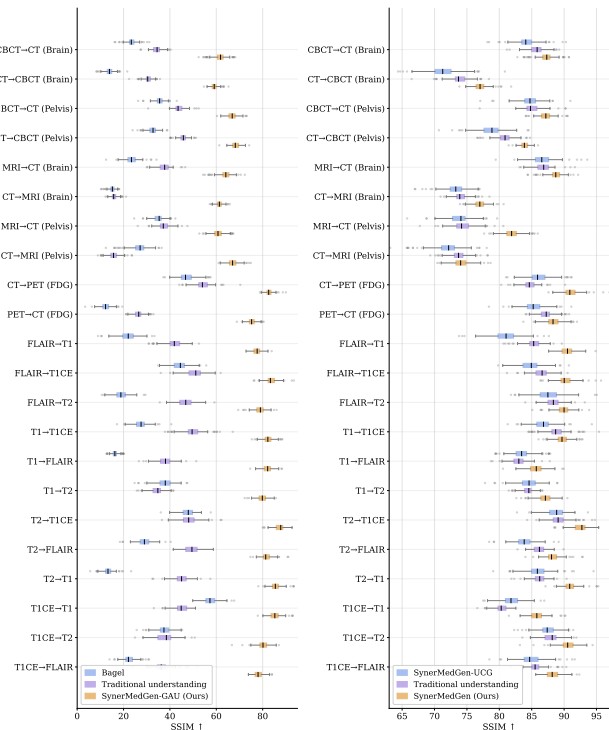

*Figure 4.* Comparison between our generation-aligned understanding and traditional understanding across 22 image synthesis tasks. Left: comparison after stage I; right: comparison after stage II.

Gen improves SSIM by 4.3% over SynDiff. Additional qualitative comparisons are provided in Appendix B.1.

**Comparison on understanding tasks.** To verify that SynerMedGen improves synthesis by acquiring *generation-relevant* understanding rather than benefiting from unrelated factors, we evaluate multiple models on the three generation-aligned understanding tasks used in stage I (GAU), namely CTS, MI, and TIA. Figure 3 reports the task accuracies of Bagel, HealthGPT, UniMedVL, and SynerMedGen. SynerMedGen consistently achieves the best performance across all three tasks, with a clear advantage in average accuracy over prior unified baselines. These results support our design rationale. The proposed tasks target the information required for conditional synthesis, and SynerMedGen explicitly learns these competencies. In contrast, existing models trained with more generic understanding objectives perform noticeably worse under the same generation-aligned evaluation. Together with the synthesis results, this evidence indicates that SynerMedGen's gains are driven by improved shared representations that directly benefit generation. We further evaluate SynerMedGen on standard medical VQA benchmarks to verify that generation-aligned understanding does not compromise general medical VQA ability; detailed results are provided in Appendix B.2.

*Table 3.* Ablation evaluation of synthesis results on the SynthRAD2023 and AutoPET datasets with SSIM (PSNR, MAE in Appendix B.1).

| Method | Schedule | Brain | | Pelvis | | Brain | | Pelvis | | Whole-Body | |
|---|---|---|---|---|---|---|---|---|---|---|---|
| | | CBCT→CT | CT→CBCT | CBCT→CT | CT→CBCT | MRI→CT | CT→MRI | MRI→CT | CT→MRI | CT→PET | PET→CT |
| Bagel (Deng et al., 2025) | Baseline | 23.37 | 13.98 | 35.59 | 32.59 | 23.34 | 15.05 | 35.24 | 27.46 | 46.87 | 12.19 |
| SynerMedGen-GAU | Stage I | **61.46**$_{+38.09}$ | **59.17**$_{+45.19}$ | **66.77**$_{+31.18}$ | **68.31**$_{+35.72}$ | **64.38**$_{+41.04}$ | **61.45**$_{+46.40}$ | **60.89**$_{+25.65}$ | **67.37**$_{+39.91}$ | **82.86**$_{+35.99}$ | **75.13**$_{+62.94}$ |
| SynerMedGen-UCG | Stage II | 84.10 | 66.36 | 84.76 | 78.75 | 86.51 | 73.10 | 73.99 | 72.36 | 85.97 | 85.19 |
| SynerMedGen | Stage I+II | **87.15**$_{+3.05}$ | **73.08**$_{+6.72}$ | **87.14**$_{+2.38}$ | **83.91**$_{+5.16}$ | **88.87**$_{+2.36}$ | **77.09**$_{+3.99}$ | **81.98**$_{+7.99}$ | **74.22**$_{+1.86}$ | **91.10**$_{+5.13}$ | **88.22**$_{+3.03}$ |

*Table 4.* Ablation of image synthesis results based on the BraTS dataset with SSIM (PSNR, MAE in Appendix B.1).

| Method | Schedule | T1→T2 | T2→T1 | T1→T1C. | T1C.→T1 | T1→FL. | FL.→T1 | T2→T1C. | T1C.→T2 | T2→FL. | FL.→T2 | T1C.→FL. | FL.→T1C. |
|---|---|---|---|---|---|---|---|---|---|---|---|---|---|
| Bagel (Deng et al., 2025) | Baseline | 38.19 | 65.44 | 27.76 | 57.52 | 16.29 | 22.58 | 47.78 | 37.37 | 28.86 | 19.37 | 22.19 | 44.11 |
| SynerMedGen-GAU | Stage I | **79.91**$_{+41.72}$ | **84.83**$_{+19.39}$ | **82.16**$_{+54.40}$ | **84.96**$_{+27.44}$ | **81.94**$_{+65.65}$ | **77.79**$_{+55.21}$ | **87.23**$_{+39.45}$ | **79.86**$_{+42.49}$ | **80.88**$_{+52.02}$ | **78.85**$_{+59.48}$ | **78.38**$_{+56.19}$ | **83.44**$_{+39.33}$ |
| SynerMedGen-UCG | Stage II | 84.68 | 85.89 | 86.93 | 81.93 | 83.48 | 81.34 | 88.78 | 87.37 | 83.78 | 87.86 | 84.75 | 87.79 |
| SynerMedGen | Stage I+II | **87.14**$_{+2.46}$ | **90.58**$_{+4.69}$ | **89.64**$_{+2.71}$ | **85.65**$_{+3.72}$ | **85.58**$_{+2.10}$ | **90.62**$_{+9.28}$ | **92.45**$_{+3.67}$ | **90.41**$_{+3.04}$ | **87.85**$_{+4.07}$ | **89.92**$_{+2.06}$ | **88.39**$_{+3.64}$ | **90.08**$_{+2.29}$ |

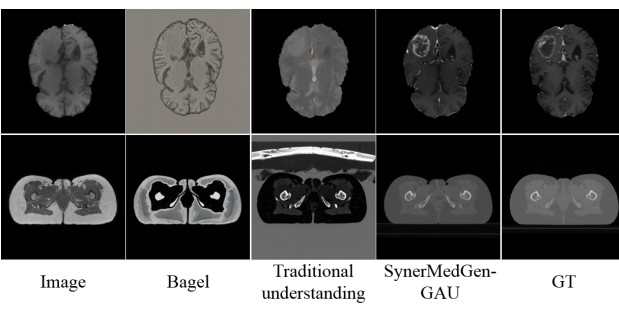

| Image | Bagel | Traditional understanding | SynerMedGen-GAU | GT |

*Figure 5.* Visual comparison of synthesized images between our generation-aligned understanding and traditional understanding.

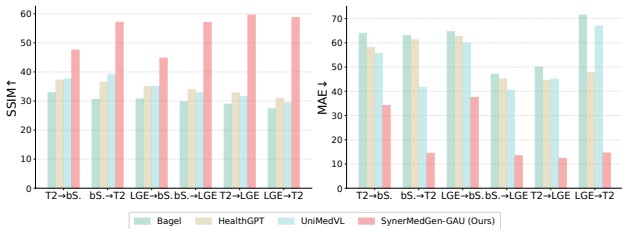

*Figure 6.* Comparison of the generalization performance of different methods on the unseen MyoPS cardiac MRI dataset.

### 4.3. Importance of Task Alignment for Generation

To validate that our cross-modality image synthesis improvements are indeed attributable to our designed generation-aligned understanding tasks, we perform a strictly controlled comparison. Specifically, we change only our generation-aligned understanding tasks to traditional medical understanding tasks from HealthGPT, while keeping the same network architecture, paired synthesis data, optimization schedule, and stage II training.

**Comparison after stage I without training generation branch.** We first compare the models' performance after the training of stage I. As shown in Figure 4 (left), traditional understanding tasks yield only limited and inconsistent improvements after stage I training. In contrast, our generation-aligned understanding consistently and substantially outperforms the baseline and traditional understanding across all 22 synthesis tasks. Notably, adding our understanding tasks alone without training the generation branch already obtains substantial improvements on the image synthesis performance, , which can be regarded as a zero-shot scenario, demonstrating that representations learned through generation-aligned understanding tasks are highly beneficial for cross-modality image synthesis.

**Comparison after stage II with complete training.** We then compare the models' performance after both stage I and stage II training. As shown in Figure 4 (right), us-

ing traditional understanding tasks in stage I obtains only marginal gains over the baseline, and can even degrade performance on certain synthesis tasks, e.g., T1→FLAIR. In contrast, initializing stage II from our GAU-based stage I training consistently improves the final performance. Overall, these results validate our key insight that in unified medical MLLMs, aligning understanding tasks with generation tasks is important to enhance the generation performance.

**Qualitative results.** The synthesized images in Figure 5 also show that models trained with traditional understanding tasks present pronounced hallucinations, while our GAU benefits image synthesis to generate outputs that better match the target-modality appearance and effectively suppress spurious artifacts (additional visual comparisons are provided in Appendix C.1.).

### 4.4. Ablation Analysis of Our Method

We assess the impact of generation-aligned understanding under the following experimental settings: (i) baseline model without training (ii) training only the understanding branch (stage I only, denoted as **SynerMedGen-GAU**), (iii), training only the generation branch (stage II only, denoted as **SynerMedGen-UCG**), and (iv) training both the understanding branch and generation branch (stage I and stage II, denoted as **SynerMedGen**).

**Effectiveness of each key component.** We first compare SynerMedGen-GAU with the Bagel baseline. As reported in Table 3 and Table 4, GAU alone already yields sub-

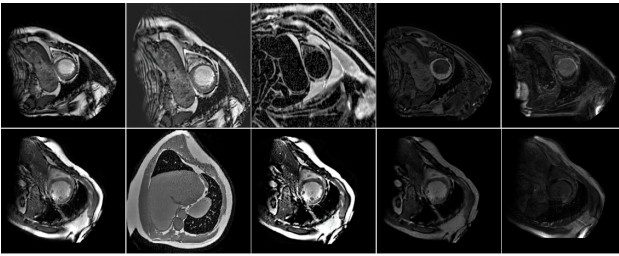

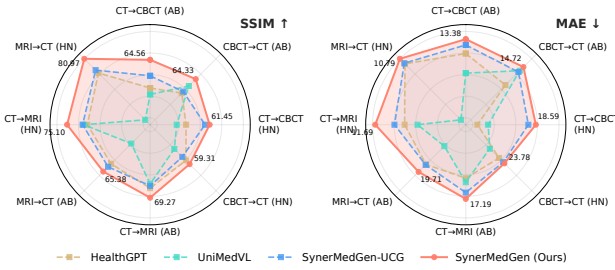

Input (bSSFP)   HealthGPT   UniMedVL   SynerMedGen-GAU   GT (LGE)

*Figure 7.* Zero-shot image synthesis comparison of different methods on the unseen MyoPS cardiac MRI dataset.

*Figure 8.* Comparison of the generalization performance of different methods on the unseen SynthRAD2025 dataset.

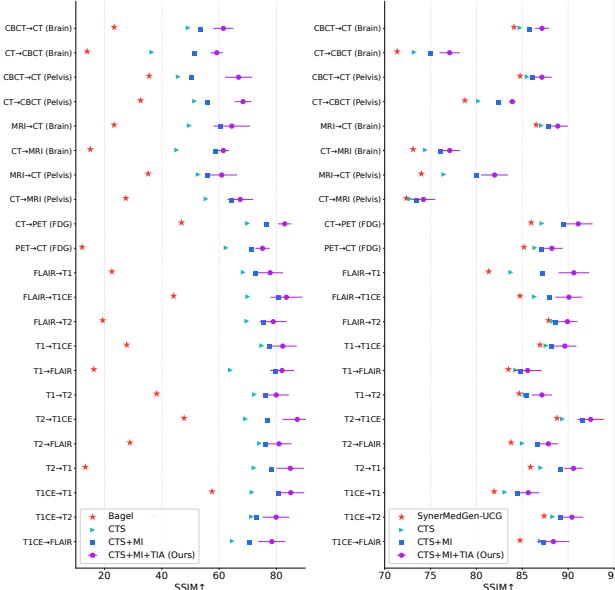

*Figure 9.* Ablation on each generation-aligned understanding task (CTS, CTS+MI, CTS+MI+TIA). Left: comparison after stage I; Right: comparison after stage II.

stantial gains on image synthesis across modalities and datasets, even though no specific generation training is performed. For example, SynerMedGen-GAU improves SSIM by 41.72% on BraTS T1→T2 and by 62.94% on whole-body PET→CT. Qualitatively, SynerMedGen-GAU reduces the hallucinated structures commonly observed in baseline outputs (see Appendix C.1 for additional examples). These results suggest that generation-aligned understanding training learns transferable representations for image synthesis. We then compare the final model SynerMedGen with SynerMedGen-UCG trained without GAU. Incorporating GAU consistently improves image synthesis performance (Table 3, Table 4). For instance, SynerMedGen yields 7.99% and 9.28% SSIM gains on Pelvis MRI→CT and BraTS FLAIR→T1, respectively.

**Generalizability of SynerMedGen-GAU.** We further evaluate the generalization performance of our SynerMedGen-GAU in comparison with Bagel, HealthGPT, and UniMedVL on the unseen MyoPS cardiac multi-sequence MRI dataset (Li et al., 2023c), which is not used in the training pipelines of any of the evaluated methods, including ours. As presented in Figure 6, our SynerMedGen-GAU consistently achieves substantially better SSIM and MAE than Bagel, HealthGPT, and UniMedVL across all evaluated image synthesis tasks. Qualitative results in Figure 7 further show that for bSSFP→LGE, our predictions better match the target-modality appearance while effectively suppressing spurious artifacts. These results indicate that the proposed generation-aligned understanding strategy substantially improves generalization to unseen imaging modalities.

**Generalizability of SynerMedGen.** We also evaluate the generalization performance of our full model SynerMedGen in comparison with SynerMedGen-UCG, HealthGPT, and UniMedVL on the unseen SynthRAD2025 dataset, which presents different organs compared to SynthRAD2023 while sharing the same modalities. As shown in Figure 8, SynerMedGen outperforms SynerMedGen-UCG, HealthGPT, and UniMedVL on both SSIM and MAE, indicating that our generation-aligned understanding provides cross-modal priors that remain effective during subsequent generation training and benefit previously unseen domains.

**Effect of each generation-aligned understanding task.** We ablate the three generation-aligned understanding tasks used in stage I, i.e., **CTS**, **MI**, and **TIA**, to quantify their individual contributions and the benefit of combining them. Following the training schedule, we progressively add these tasks during stage I in the order CTS → CTS+MI → CTS+MI+TIA. We evaluate each variant both after stage I training (Figure 9 (left)) and after stage I and stage II training (Figure 9 (right)). Starting from the baseline, adding CTS consistently improves image synthesis performance, indicating that correspondence-focused understanding transfers effectively to cross-modality image synthesis. Adding MI further enhances performance, and the full combination with TIA achieves the best overall results. The consistent gains across the progressive variants suggest that the three tasks provide complementary benefits and together form a more effective understanding training strategy. Beyond the progressive ablation, we provide all individual and paired task combinations after both stage I and stage I+II training in Appendix B.3, which further confirms the complementary

contribution of CTS, MI, and TIA.

## 5. Conclusion

We found that the bottleneck in models for medical multimodal understanding and generation lies in the gap between understanding and generation. To bridge this gap, we propose SynerMedGen, which extracts understanding-oriented supervision from paired generative data via three understanding tasks (CTS/MI/TIA). Across different multimodal generation tasks, SynerMedGen consistently improves generation quality and reduces hallucinations, indicating that generation-aligned understanding is a key lever for unified medical image generation. While the current framework is developed and evaluated on 2D medical images, future work will extend SYNERMEDGEN to 3D volumetric modeling to better exploit spatial anatomical context in modalities such as CT and MRI. We will also explore broader medical generation tasks, additional imaging modalities, and the integration of structured clinical knowledge to further narrow the gap between understanding and generation.

## Acknowledgements

The work described in this paper is supported by grants from HKU Startup Fund and HKU Seed Fund for Basic Research.

## Impact Statement

This paper presents work whose goal is to advance the field of Machine Learning. There are many potential societal consequences of our work, none which we feel must be specifically highlighted here.

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

## A. Dataset Description

**SynerMed.** We build *SynerMed*, a large-scale paired corpus for cross-modality medical image synthesis. As shown in Figure 10a, it integrates multiple public datasets spanning CBCT, CT, PET, and multi-sequence MRI (e.g., T1/T1CE/T2/FLAIR), and covers diverse anatomies. We define **22** directed synthesis tasks across these modalities and curate approximately **1M** paired samples. Each sample is a *slice-aligned* pair $(x_{\text{src}}, x_{\text{tgt}})$ associated with a target-modality request, providing the supervision needed for medical images synthesis.

**Generation-aligned understanding supervision.** From the same paired corpus, we derive *generation-aligned* vision–language supervision for stage I (GAU) in the main paper. Concretely, we construct three task families: **Conditional Target Selection (CTS)** for slice-level correspondence under a target-modality condition, **Modality Identification (MI)** for explicit modality controllability, and **Transformation Instruction Alignment (TIA)** for grounding mapping direction and constraints in language. This results in **2.3M** prompted understanding instances, with the task distribution shown in Figure 10b. Together, the paired synthesis data and the derived understanding tasks form a unified testbed for studying synergy between multimodal understanding and conditional medical image generation.

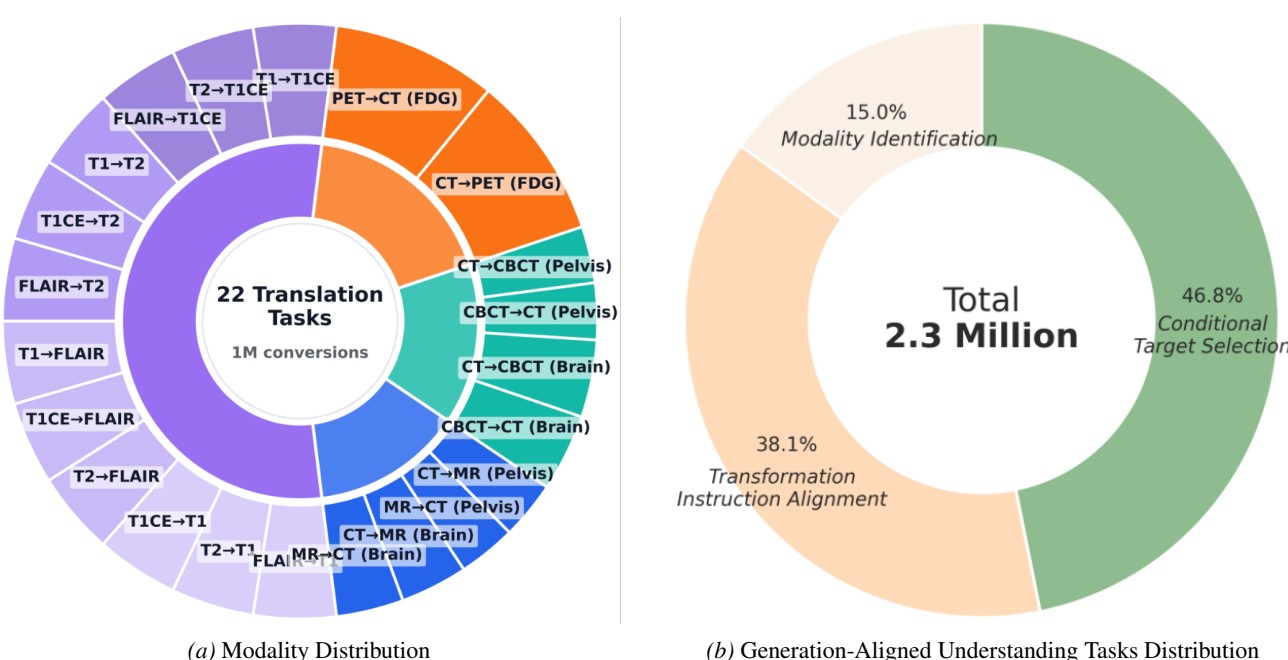

*(a)* Modality Distribution           *(b)* Generation-Aligned Understanding Tasks Distribution

*Figure 10.* Comprehensive Statistical Analysis of the Curated Multimodal Synthetic Medical Dataset and Our Designed Generation-Aligned Understanding Tasks.

## B. Supplementary Experimental Results

This section presents additional experimental results that support the findings discussed in the main paper. These results are organized into several subsections, each highlighting different aspects of the model performance.

### B.1. Quantitative Comparison with Baseline Methods

We report additional quantitative results using **MAE** and **PSNR**. Table 5 and Table 6 summarize MAE on the multi-dataset benchmark and BraTS, respectively. SYNERMEDGEN achieves the lowest MAE on most tasks, spanning CBCT↔CT, CT↔MRI, and PET↔CT (e.g., Brain CT→CBCT: 34.29; Whole-Body CT→PET: 1.57), and remains competitive on the remaining tasks. Table 7 and Table 8 report PSNR. Consistent with the MAE trends, SYNERMEDGEN achieves the highest PSNR on most tasks, indicating improved reconstruction fidelity across both multi-organ synthesis and multi-sequence MRI translation.

*Table 5.* Quantitative comparison of image synthesis methods on SynthRAD2023 and AutoPET datasets with MAE↓.

| Method | Brain | | Pelvis | | Brain | | Pelvis | | Whole-Body | |
|---|---|---|---|---|---|---|---|---|---|---|
| | CBCT→CT | CT→CBCT | CBCT→CT | CT→CBCT | MRI→CT | CT→MRI | MRI→CT | CT→MRI | CT→PET | PET→CT |
| Bagel (Deng et al., 2025) | 71.74 | 90.80 | 64.63 | 72.47 | 82.81 | 81.63 | 70.25 | 70.38 | 50.04 | 88.47 |
| SynerMedGen-GAU | 52.09 | 47.84 | 28.79 | 39.18 | 40.82 | 42.01 | 45.53 | 50.79 | 7.13 | 7.84 |
| Pix2Pix (Isola et al., 2017) | 35.65 | 44.40 | 29.26 | 43.83 | 31.01 | 40.13 | 39.72 | 59.85 | 11.98 | 9.95 |
| CycleGAN (Zhu et al., 2017) | 43.34 | 59.09 | 37.99 | 49.12 | 50.55 | 45.16 | 56.72 | 64.30 | 16.92 | 17.33 |
| BBDM (Li et al., 2023a) | 27.12 | 37.06 | 32.35 | 40.12 | 28.12 | 42.41 | 36.16 | 67.77 | 3.39 | 4.05 |
| ResViT (Dalmaz et al., 2022) | 22.51 | 35.78 | 19.94 | 34.93 | 24.56 | 35.97 | 30.36 | 48.96 | 2.06 | 1.93 |
| SynDiff (Özbey et al., 2023) | 21.75 | 36.50 | 21.88 | 32.99 | 23.21 | 35.91 | 29.74 | 47.83 | 3.56 | 3.34 |
| RCD (Li et al., 2025) | 21.30 | 35.99 | 20.01 | **31.51** | 24.77 | 34.71 | 30.28 | 45.58 | 3.17 | 3.98 |
| HealthGPT (Lin et al., 2025) | 44.82 | 67.73 | 45.65 | 46.39 | 27.88 | 33.63 | 28.31 | 43.29 | 26.33 | 22.85 |
| UniMedVL (Ning et al., 2025) | 40.12 | 77.41 | 44.09 | 47.15 | 44.55 | 38.78 | 49.10 | 57.85 | 17.15 | 26.23 |
| SynerMedGen-UCG | 23.81 | 36.05 | 22.76 | 33.97 | 25.40 | 37.25 | 31.60 | 47.12 | 2.49 | 2.36 |
| SynerMedGen | **20.21**$_{-3.60}$ | **34.29**$_{-1.76}$ | **19.31**$_{-3.45}$ | 31.66$_{-2.31}$ | **23.54**$_{-1.86}$ | **32.71**$_{-4.54}$ | **29.51**$_{-2.09}$ | **44.98**$_{-2.14}$ | **1.57**$_{-0.92}$ | **1.81**$_{-0.55}$ |

*Table 6.* Quantitative comparison of image synthesis methods on BraTS dataset with MAE↓.

| Method | T1→T2 | T2→T1 | T1→T1C. | T1C.→T1 | T1→FL. | FL.→T1 | T2→T1C. | T1C.→T2 | T2→FL. | FL.→T2 | T1C.→FL. | FL.→T1C. |
|---|---|---|---|---|---|---|---|---|---|---|---|---|
| Bagel (Deng et al., 2025) | 38.77 | 28.42 | 72.28 | 46.70 | 98.35 | 97.65 | 54.50 | 45.00 | 91.04 | 93.70 | 91.80 | 45.26 |
| SynerMedGen-GAU | 7.49 | 12.26 | 4.93 | 9.39 | 9.19 | 19.82 | 3.96 | 8.52 | 12.13 | 9.60 | 15.31 | 5.74 |
| Pix2Pix (Isola et al., 2017) | 17.54 | 21.91 | 16.96 | 17.60 | 17.66 | 11.64 | 12.48 | 17.62 | 15.07 | 19.36 | 16.49 | 13.82 |
| CycleGAN (Zhu et al., 2017) | 19.71 | 24.86 | 26.98 | 24.00 | 29.24 | 17.14 | 24.64 | 25.79 | 22.40 | 21.75 | 31.46 | 26.77 |
| BBDM (Li et al., 2023a) | 7.51 | 6.58 | 4.90 | 11.16 | 8.01 | 6.32 | 5.01 | 7.27 | 6.91 | 6.29 | 7.43 | 5.57 |
| ResViT (Dalmaz et al., 2022) | 5.46 | 6.38 | 4.91 | 6.59 | 5.63 | 7.24 | 3.80 | 4.48 | 5.49 | 4.99 | 5.76 | 4.80 |
| SynDiff (Özbey et al., 2023) | 5.71 | 5.25 | **3.57** | 5.43 | 6.72 | 6.04 | 3.67 | 4.90 | 5.10 | 4.27 | 5.53 | 3,97 |
| RCD (Li et al., 2025) | 5.12 | 6.09 | 4.47 | 6.11 | 5.94 | 5.63 | 4.35 | 5.15 | 6.55 | 3.91 | 5.25 | 4.50 |
| HealthGPT (Lin et al., 2025) | 14.15 | 19.65 | 17.92 | 17.19 | 17.83 | 23.12 | 19.65 | 13.79 | 19.37 | 17.07 | 20.03 | 19.30 |
| UniMedVL (Ning et al., 2025) | 9,90 | 12.59 | 15.14 | 14.23 | 10.83 | 18.58 | 16.81 | 9.06 | 14.53 | 13.79 | 11.00 | 15.65 |
| SynerMedGen-UCG | 5.97 | 11.59 | 6.07 | 10.44 | 6.85 | 7.55 | 3.30 | 5.21 | 8.02 | 5.90 | 8.01 | 4.71 |
| SynerMedGen | **4.51**$_{-1.46}$ | **4.52**$_{-7.07}$ | 3.86$_{-2.21}$ | **5.21**$_{-5.23}$ | **5.07**$_{-1.78}$ | **5.04**$_{-2.51}$ | **3.05**$_{-0.25}$ | **4.39**$_{-0.82}$ | **4.46**$_{-3.56}$ | **3.59**$_{-2.31}$ | **4.49**$_{-3.52}$ | **3.22**$_{-1.49}$ |

*Table 7.* Quantitative comparison of image synthesis methods on SynthRAD2023 and AutoPET datasets with PSNR↑.

| Method | Brain | | Pelvis | | Brain | | Pelvis | | Whole-Body | |
|---|---|---|---|---|---|---|---|---|---|---|
| | CBCT→CT | CT→CBCT | CBCT→CT | CT→CBCT | MRI→CT | CT→MRI | MRI→CT | CT→MRI | CT→PET | PET→CT |
| Bagel (Deng et al., 2025) | 9.63 | 7.40 | 9.57 | 11.34 | 8.93 | 5.97 | 11.62 | 7.32 | 9.89 | 6.03 |
| SynerMedGen-GAU | 28.90 | 20.52 | 28.68 | 22.94 | 27.15 | 26.24 | 27.42 | 27.70 | 24.42 | 25.65 |
| Pix2Pix (Isola et al., 2017) | 33.18 | 23.76 | 32.02 | 24.73 | 33.09 | 31.10 | 31.17 | 30.40 | 19.21 | 20.77 |
| CycleGAN (Zhu et al., 2017) | 32.45 | 22.31 | 31.71 | 23.15 | 30.77 | 30.79 | 28.10 | 29.12 | 17.72 | 18.03 |
| BBDM (Li et al., 2023a) | 33.25 | 24.83 | 32.21 | 24.97 | 32.23 | 31.67 | 32.15 | 30.41 | 24.17 | 22.50 |
| ResViT (Dalmaz et al., 2022) | 34.12 | 26.33 | 25.90 | 25.88 | 33.72 | 32.76 | 32.91 | 30.78 | 28.84 | 29.12 |
| SynDiff (Özbey et al., 2023) | 34.20 | 26.46 | 33.82 | 26.14 | 33.98 | 32.60 | 33.07 | 31.06 | **30.27** | 29.17 |
| RCD (Li et al., 2025) | 34.21 | 26.13 | 33.67 | 26.36 | 33.79 | 32.31 | 32.88 | 31.23 | 28.77 | 28.82 |
| HealthGPT (Lin et al., 2025) | 21.31 | 17.92 | 19.21 | 17.24 | 33.97 | 32.58 | 33.46 | 30.71 | 14.21 | 18.59 |
| UniMedVL (Ning et al., 2025) | 19.09 | 15.88 | 19.79 | 18.30 | 20.74 | 15.95 | 19.23 | 16.01 | 17.45 | 16.96 |
| SynerMedGen-UCG | 34.17 | 25.36 | 33.72 | 18.75 | 32.15 | 33.28 | 33.34 | 30.39 | 25.94 | 28.24 |
| SynerMedGen | **35.03**$_{+0.86}$ | **26.97**$_{+1.61}$ | **34.51**$_{+0.79}$ | **23.32**$_{+4.57}$ | **34.13**$_{+1.98}$ | **33.87**$_{+0.59}$ | **34.52**$_{+1.18}$ | **31.65**$_{+1.26}$ | 30.08$_{+4.14}$ | **30.09**$_{+1.85}$ |

*Table 8.* Quantitative comparison of image synthesis methods on BraTS dataset with PSNR↑.

| Method | T1→T2 | T2→T1 | T1→T1C. | T1C.→T1 | T1→FL. | FL.→T1 | T2→T1C. | T1C.→T2 | T2→FL. | FL.→T2 | T1C.→FL. | FL.→T1C. |
|---|---|---|---|---|---|---|---|---|---|---|---|---|
| Bagel (Deng et al., 2025) | 12.89 | 13.31 | 10.03 | 11.97 | 4.22 | 6.95 | 12.98 | 13.88 | 6.52 | 5.65 | 3.75 | 13.35 |
| SynerMedGen-GAU | 20.36 | 18.98 | 25.92 | 19.87 | 19.54 | 18.36 | 24.93 | 20.16 | 17.32 | 18.79 | 17.12 | 24.70 |
| Pix2Pix (Isola et al., 2017) | 18.57 | 15.72 | 18.94 | 15.16 | 18.41 | 17.91 | 19.14 | 18.97 | 15.56 | 18.36 | 17.42 | 18.33 |
| CycleGAN (Zhu et al., 2017) | 17.20 | 15.47 | 16.12 | 13.55 | 15.03 | 14.67 | 17.88 | 17.98 | 15.63 | 15.27 | 13.28 | 17.69 |
| BBDM (Li et al., 2023a) | 21.33 | 22.53 | 26.38 | 19.35 | 21.27 | 23.46 | 26.85 | 21.55 | 22.04 | 22.54 | 21.29 | 26.30 |
| ResViT (Dalmaz et al., 2022) | 22.80 | 22.81 | 25.60 | 22.25 | 23.35 | 21.97 | 27.13 | 25.87 | 23.67 | 24.53 | 23.14 | 25.74 |
| SynDiff (Özbey et al., 2023) | 22.17 | 24.51 | 27.87 | 23.30 | 23.01 | 24.10 | 28.63 | 25.05 | 24.27 | 24.88 | 24.29 | 27.39 |
| RCD (Li et al., 2025) | 23.07 | 24.85 | 27.24 | 22.31 | 23.45 | 24.39 | 27.63 | 25.39 | 23.49 | 25.31 | 23.33 | 27.01 |
| HealthGPT (Lin et al., 2025) | 17.66 | 15.23 | 15.54 | 14.22 | 16.72 | 15.26 | 16.98 | 17.86 | 15.54 | 16.12 | 16.60 | 15.99 |
| UniMedVL (Ning et al., 2025) | 19.38 | 16.74 | 17.67 | 16.18 | 17.94 | 15.42 | 16.64 | 19.52 | 15.77 | 16.99 | 17.70 | 17.02 |
| SynerMedGen-UCG | 22.07 | 18.34 | 24.75 | 19.54 | 22.27 | 21.72 | 27.93 | 23.99 | 20.49 | 22.56 | 20.68 | 27.16 |
| SynerMedGen | **24.93**$_{+2.86}$ | **25.53**$_{+7.19}$ | **29.08**$_{+4.33}$ | **24.89**$_{+5.35}$ | **24.59**$_{+2.32}$ | **25.14**$_{+3.42}$ | **30.11**$_{+2.18}$ | **27.02**$_{+3.03}$ | **26.18**$_{+5.69}$ | **26.41**$_{+3.85}$ | **25.03**$_{+4.35}$ | **28.76**$_{+1.60}$ |

*Table 9.* Results on standard medical VQA benchmarks.

| Method | VQA-RAD | SLAKE | OmniMedVQA | MMMU-Med |
|---|---|---|---|---|
| InternVL2 | 49.0 | 50.1 | 54.5 | 43.3 |
| Bagel | 60.1 | 58.9 | 71.1 | 46.5 |
| HealthGPT | 58.3 | 64.5 | 74.4 | 49.2 |
| SynerMedGen | **61.3** | **69.4** | **78.9** | **50.1** |

*Table 10.* Individual and paired task ablations evaluated after Stage I and Stage I+II training. The baseline is Bagel for Stage I and SynerMedGen-UCG for Stage I+II.

| Method | Stage I SSIM | Stage I+II SSIM |
|---|---|---|
| Baseline | 29.14 | 82.62 |
| CTS | 61.83 | 83.79 |
| MI | 50.72 | 83.34 |
| TIA | 61.86 | 84.40 |
| CTS+MI | 68.92 | 85.16 |
| CTS+TIA | 69.00 | 85.77 |
| MI+TIA | 66.75 | 85.22 |
| CTS+MI+TIA (Ours) | **74.91** | **86.59** |

## B.2. Evaluation on Standard Medical VQA Benchmarks

To evaluate whether the proposed generation-aligned understanding tasks affect general medical visual question answering ability, we further evaluate SynerMedGen on four standard medical VQA benchmarks, including VQA-RAD, SLAKE, OmniMedVQA, and MMMU-Med. As shown in Table 9, SynerMedGen consistently outperforms Bagel and other representative medical vision-language models across all four benchmarks. These results indicate that our generation-aligned understanding tasks do not compromise general medical VQA performance. Instead, they may improve fine-grained visual representation learning, which is also beneficial for general medical VQA.

## B.3. Individual and Paired Task Ablations

To provide a more complete analysis of the contribution of each training task, we further conduct all individual and paired task ablations in addition to the progressive ablation reported in the main paper, i.e., CTS $\rightarrow$ CTS+MI $\rightarrow$ CTS+MI+TIA. We evaluate these variants after both Stage I and Stage I+II training. As shown in Table 10, each task contributes independently to the final performance. Using CTS, MI, or TIA alone improves over the corresponding baseline, while combining two tasks generally brings further gains. The full CTS+MI+TIA combination achieves the best SSIM under both training settings, indicating that the three tasks provide complementary supervision.

## B.4. Lesion-related Evaluation

To further examine whether the generated images preserve lesion-related information, we conduct a downstream brain tumor segmentation evaluation on BraTS. Specifically, we use 100 paired BraTS cases with a 70/10/20 train/validation/test split, and train nnU-Net under three input settings. As shown in Table 11, using only T1, T2, and FLAIR yields a DSC of 56.25. Adding the synthetic T1CE improves the DSC to 62.48, which is close to the performance obtained with real T1CE images, i.e., 64.83. These results indicate that our generated T1CE images preserve tumor-relevant cues that are useful for downstream lesion segmentation.

## B.5. Effect of $K$ in Interlayer Selection (Ablation Study)

We study the sensitivity of CTS to $K$, the half-window used to sample hard negatives from neighboring target slices (i.e., candidates are drawn from indices $k \pm 1, \ldots, k \pm K$ around the true counterpart). Table 12 reports the average zero-shot synthesis performance (SSIM↑, MSE↓). Overall, the effect of $K$ is non-monotonic. A moderate window ($K = 5$) achieves the best trade-off, improving SSIM from 74.33 ($K = 2$) to 74.91 and reducing MSE from 21.85 to 21.53. Increasing $K$ beyond this point gradually degrades performance (e.g., SSIM 74.71 / MSE 21.70 at $K = 8$), and the drop becomes more evident for $K \geq 10$ (SSIM 73.97→71.34; MSE 22.09→23.61 as $K$ grows from 10 to 20). We attribute this to overly

*Table 11.* Brain tumor segmentation results on BraTS under different input settings.

| Input Modalities | DSC |
|---|---|
| T1+T2+FLAIR | 56.25 |
| T1+T2+FLAIR+synthetic T1CE | 62.48 |
| T1+T2+FLAIR+real T1CE | **64.83** |

*Table 12.* Sensitivity of CTS to the inter-slice candidate window size $K$ (zero-shot average SSIM↑ and MSE↓).

| Model | $K = 2$ | | $K = 5$ | | $K = 8$ | | $K = 10$ | | $K = 15$ | | $K = 20$ | |
|---|---|---|---|---|---|---|---|---|---|---|---|---|
| | SSIM↑ | MSE↓ | SSIM↑ | MSE↓ | SSIM↑ | MSE↓ | SSIM↑ | MSE↓ | SSIM↑ | MSE↓ | SSIM↑ | MSE↓ |
| Average | 74.33 | 21.85 | **74.91** | **21.53** | 74.71 | 21.70 | 73.97 | 22.09 | 72.77 | 22.97 | 71.34 | 23.61 |

broad candidate sets weakening the correspondence signal: adding farther slices introduces many easy or less informative negatives (and potentially noisier matches), which reduces the pressure to learn fine-grained, slice-specific alignment. Unless otherwise stated, we set $K = 5$ in the remaining experiments.

## C. Visual Comparison

This section provides qualitative comparisons that follow the two-stage evaluation protocol in the main paper.

### C.1. Stage I (GAU-only): Zero-shot Comparison

Figure 11 and Figure 12 visualize zero-shot synthesis after **stage I (GAU)** only. We compare our GAU-pretrained model with (i) the unified baseline without medical understanding supervision (Bagel) and (ii) a representative baseline trained with traditional medical understanding supervision (HealthGPT). Across modality translation settings, both baselines frequently introduce hallucinated structures and unstable intensity patterns, suggesting insufficient preservation of correspondence-sensitive anatomy. In contrast, GAU yields more anatomically consistent outputs while better matching target-modality appearance, supporting our claim that generation-aligned supervision learns synthesis-sufficient representations.

### C.2. Stage I+II (GAU→UCG): End-to-end Comparison

Figure 13 and Figure 14 show end-to-end results after the full **GAU→UCG** pipeline. The examples demonstrate reliable cross-modality translation across anatomies (e.g., CT↔MRI, CBCT↔CT, and CT↔PET), with improved structural coherence and fewer artifacts. For multi-sequence MRI synthesis, the model preserves fine-grained, sequence-dependent contrasts while maintaining anatomical integrity, highlighting the benefit of transferring generation-aligned understanding into downstream generation.

### C.3. Comparison with Dedicated Synthesis Methods

In addition to the zero-shot and end-to-end comparisons above, we further compare SynerMedGen with representative image-to-image synthesis methods, including Pix2Pix, BBDM, and SynDiff. As shown in Figure 15, the proposed method achieves more faithful cross-modality synthesis across different datasets and anatomical regions.

## D. Examples of Text Prompts

The following shows the text prompt we used when synthesizing medical images of different modalities.

- **CT→CBCT:** Convert a non-contrast pelvic CT slice to a CBCT-like appearance. Keep anatomy, voxel grid, field-of-view, slice position, and lesion morphology strictly unchanged (1:1 mapping). Modify only photometric/texture properties. CBCT appearance constraints (ONLY IF VISIBLE): • Lower soft-tissue contrast vs diagnostic CT; HU compression acceptable. • Add realistic, moderate CBCT texture: granular noise, mild cone-beam streaks/flare, slight ring artifacts; must not hide organ boundaries. • Gentle scatter shading/cupping permissible; avoid strong vignetting or truncation that alters anatomy. Preserve stones/hardware geometry and position. No hallucinated anatomy or warping.

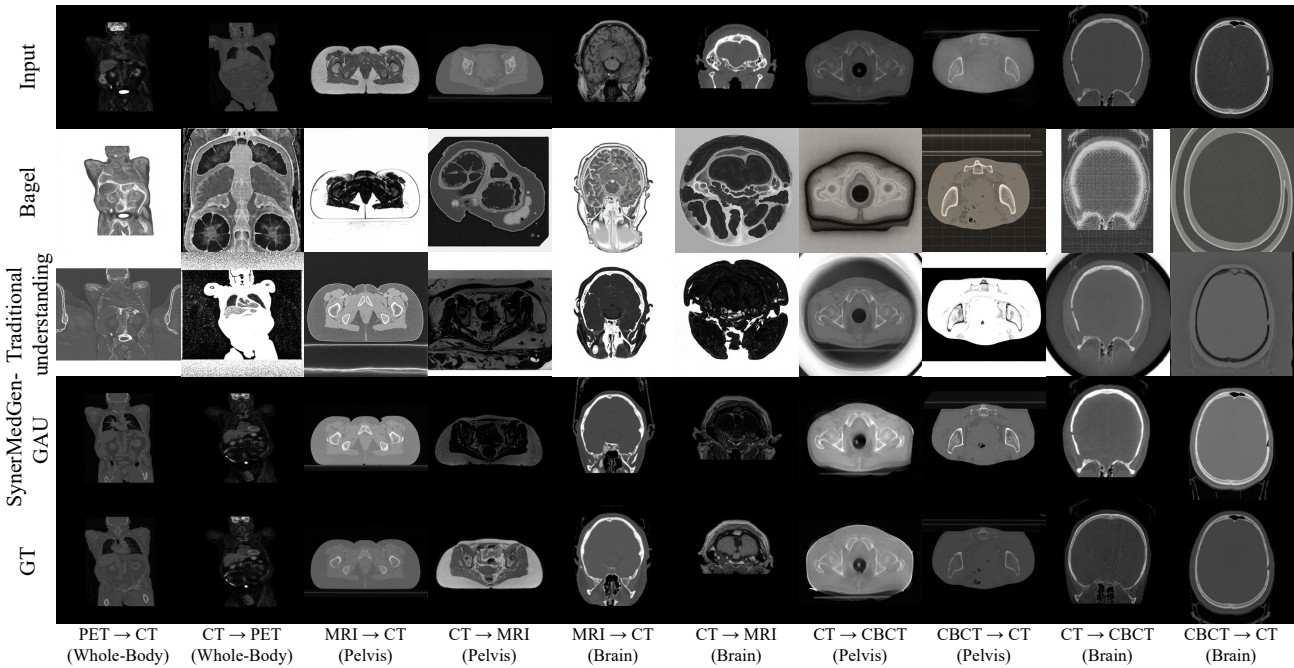

*Figure 11.* Zero-shot visual comparison of synthesized images by different methods on the SynthRAD2023 and AutoPET datasets.

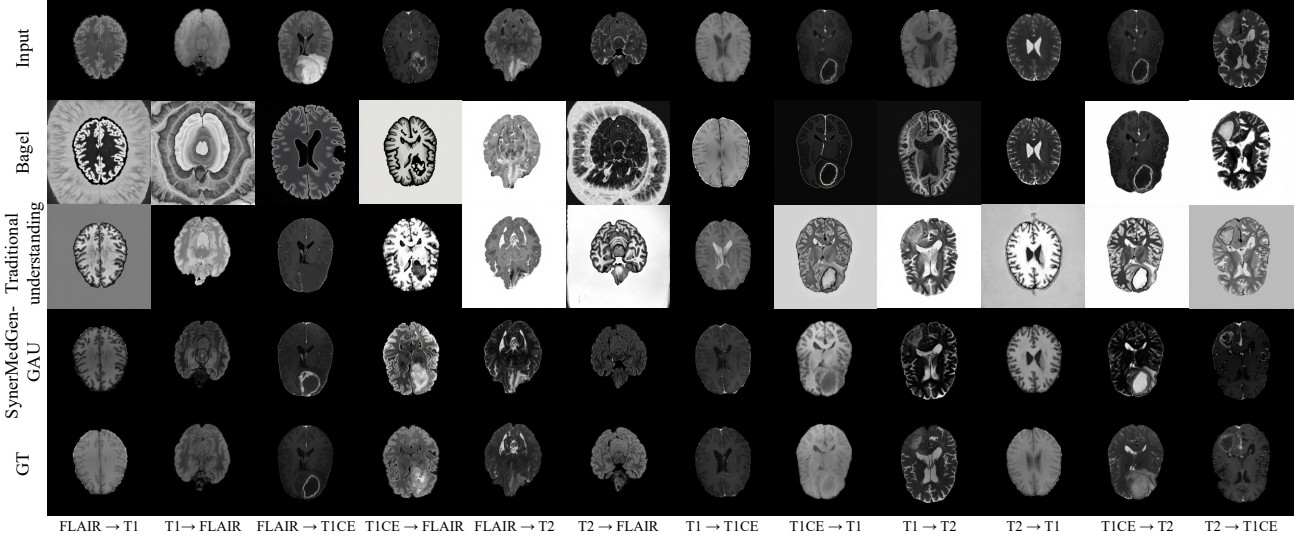

*Figure 12.* Zero-shot visual comparison of synthesized images by different methods on the BraTS dataset.

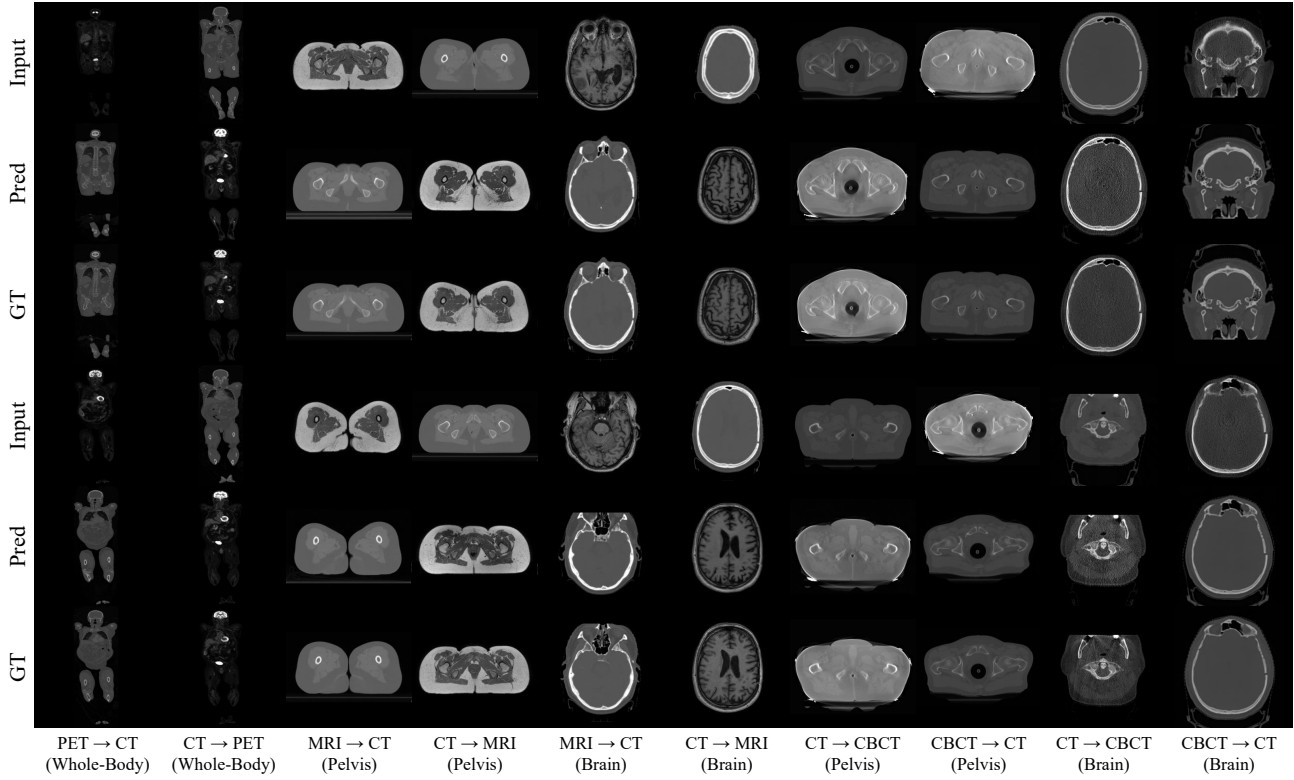

*Figure 13.* Case studies of different modalities synthesis.

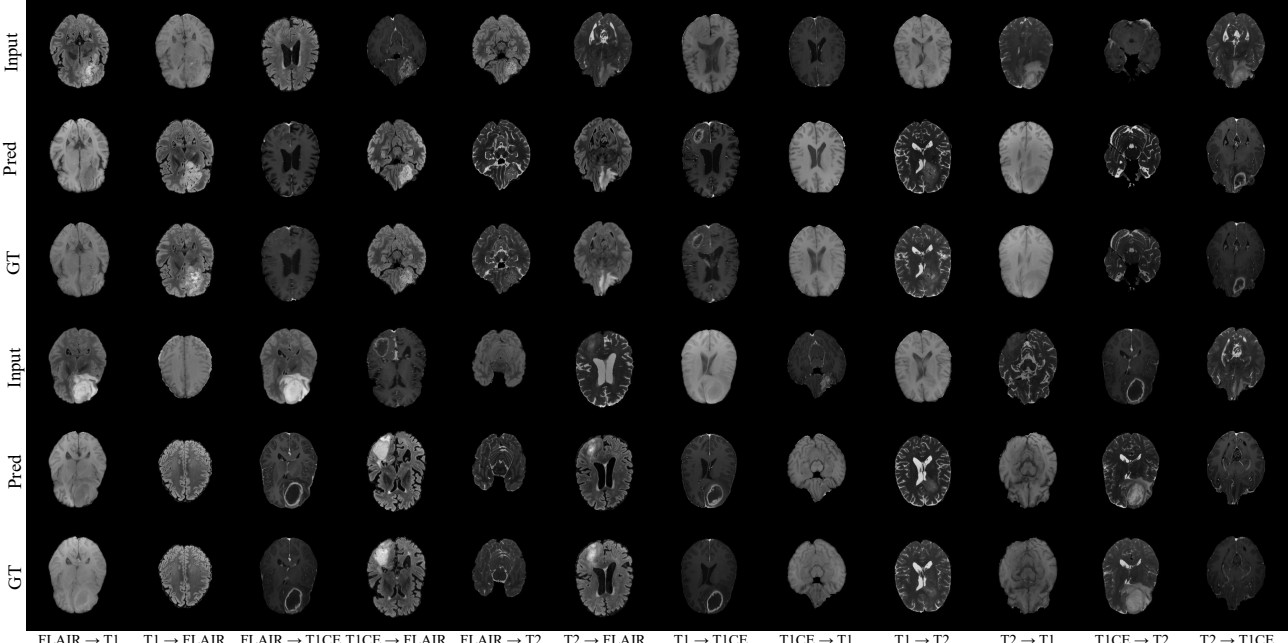

*Figure 14.* Case studies of different MRI synthesis.

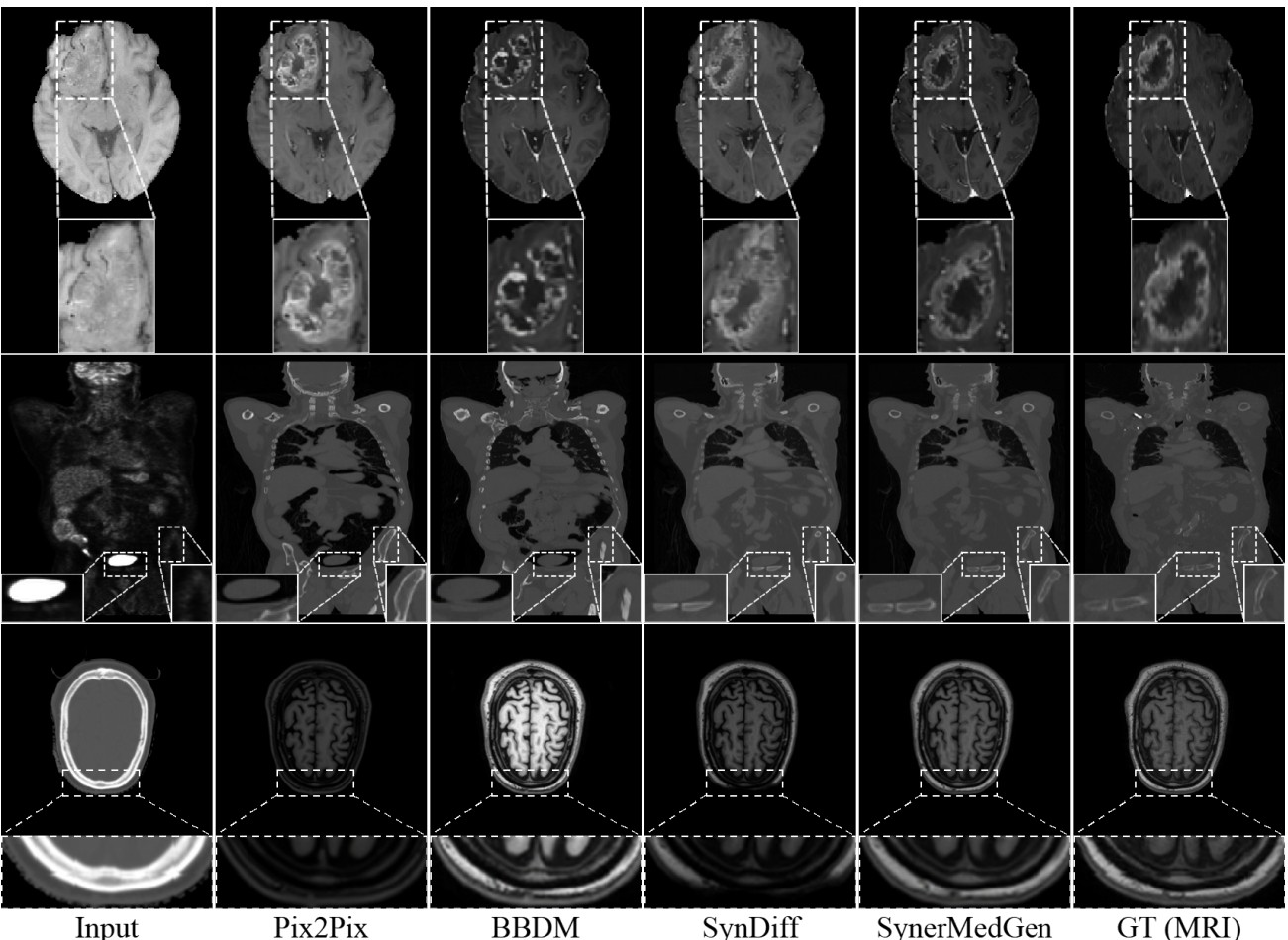

| Input | Pix2Pix | BBDM | SynDiff | SynerMedGen | GT (MRI) |

*Figure 15.* Visual comparison of different synthesis methods on the different datasets.

- **PET→CT:** Convert an PET whole-body slice to a non-contrast diagnostic CT appearance. Keep anatomy, voxel grid, field-of-view, slice position, and lesion morphology strictly unchanged (1:1 mapping). Modify ONLY contrast to emulate CT attenuation. CT constraints (ONLY IF VISIBLE): • Cortical/trabecular bone → bright to very bright; air spaces → near-black. • Soft-tissue HU ordering: fat < water (mid-gray) < muscle/solid organs < bone. • Remove PET blur/halo appearance; produce realistic diagnostic CT noise/edge sharpness. No iodinated-contrast patterns or invented anatomy; preserve exact geometry.

- **CT→MRI:** Convert a non-contrast brain CT slice to a non-contrast structural MRI (T1-like) appearance. Keep anatomy, voxel grid, field-of-view, slice position and lesion morphology strictly unchanged (1:1 mapping). Modify ONLY soft-tissue signal relationships to emulate MRI. MRI (T1-like) constraints (ONLY IF VISIBLE): • Cortical bone and air → near-black. • White matter slightly brighter than gray matter. • Ventricles/sulci → CSF dark with sharp boundaries. • Remove CT-specific streaks/beam hardening cues. No gadolinium enhancement patterns, no invented anatomy; preserve exact geometry and realistic MRI-like texture.

- **MRI T2→MRI T1CE:** Generate a post-contrast MRI FLAIR (T1-weighted Contrast-Enhanced Magnetic Resonance Imaging) depiction from the MRI FLAIR (Fluid-Attenuated Inversion Recovery (FLAIR) Magnetic Resonance Imaging) slice while rigorously preserving anatomy. Remove FLAIR-specific CSF suppression characteristics, enforce T1-like baseline (dark CSF), and add anatomically plausible enhancement (vessels, dura, genuinely enhancing tumor regions) without altering lesion size or shape. No artificial structures; maintain resolution and field-of-view.

- **MRI T2→MRI T1:** Render a faithful MRI T1 (T1-weighted Magnetic Resonance Imaging) version from the MRI T2 (T2-weighted Magnetic Resonance Imaging) slice by changing contrast only. In MRI T1, CSF should be dark; white matter typically brighter than gray matter; no contrast-agent signatures should appear. Exact geometry, field-of-view, and lesion morphology must be preserved; avoid hallucinations.

- **MRI T1→MRI T2:** Convert this MRI T1 (T1-weighted Magnetic Resonance Imaging) slice into a fluid-sensitive MRI T2 (T2-weighted Magnetic Resonance Imaging) depiction. Preserve geometry exactly; alter only signal relationships so CSF/free fluid becomes bright, vasogenic edema and many lesions trend hyperintense, and—on MRI T2—white matter appears darker than gray matter (contrast direction reversed from T1). Exclude any contrast-agent effects; no structure may be added, removed, or reshaped.

# E. Specific Generation-Aligned Understanding Tasks

This section provides a detailed presentation of three Generation-Aligned Understanding tasks: Conditional Target Selection, Modality Identification, and Transformation Instruction Alignment.

## E.1. Conditional Target Selection

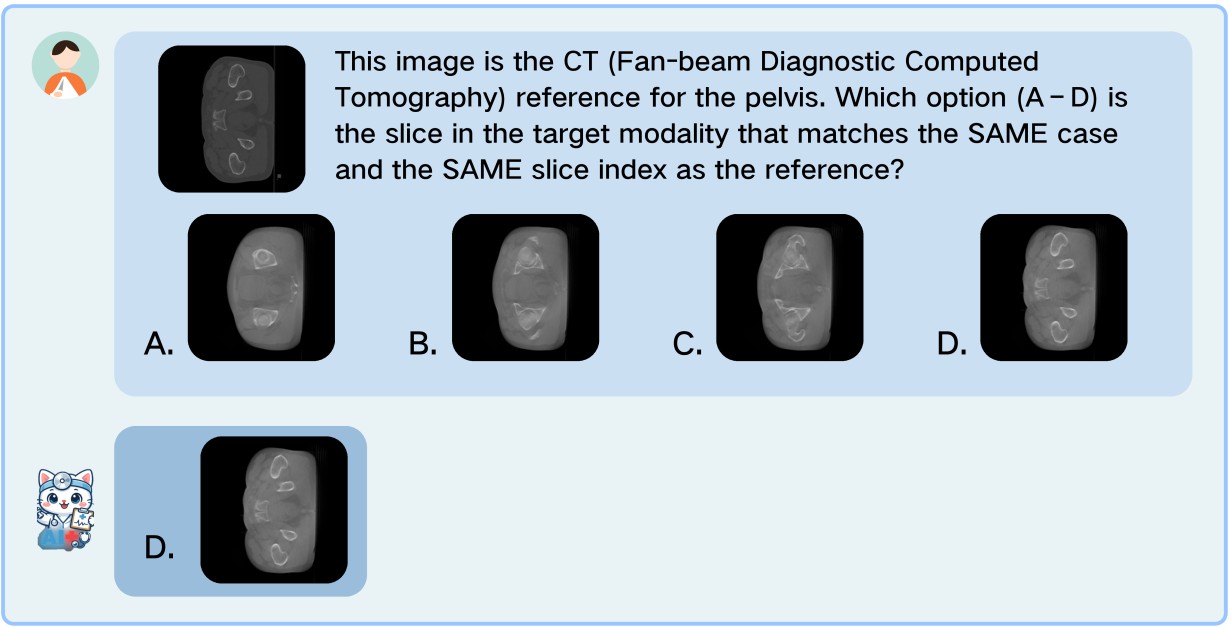

*Figure 16.* Visual question answering example demonstrating cross-modality slice alignment from CT to CBCT.

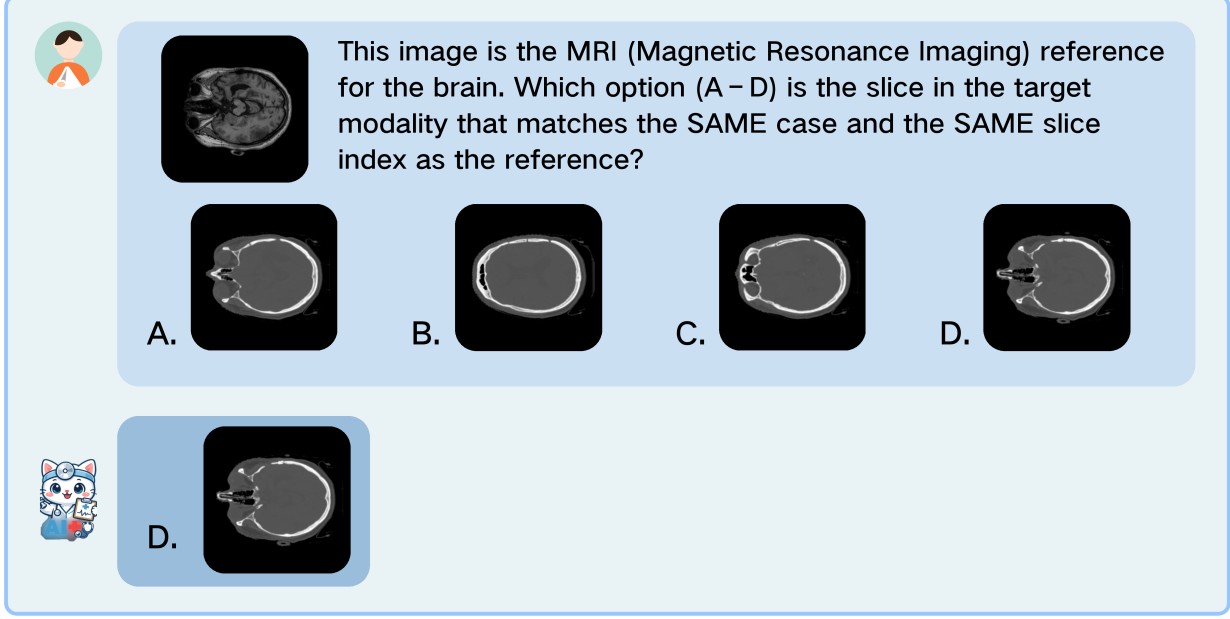

*Figure 17.* Visual question answering example demonstrating cross-modality slice alignment from MRI to CT.

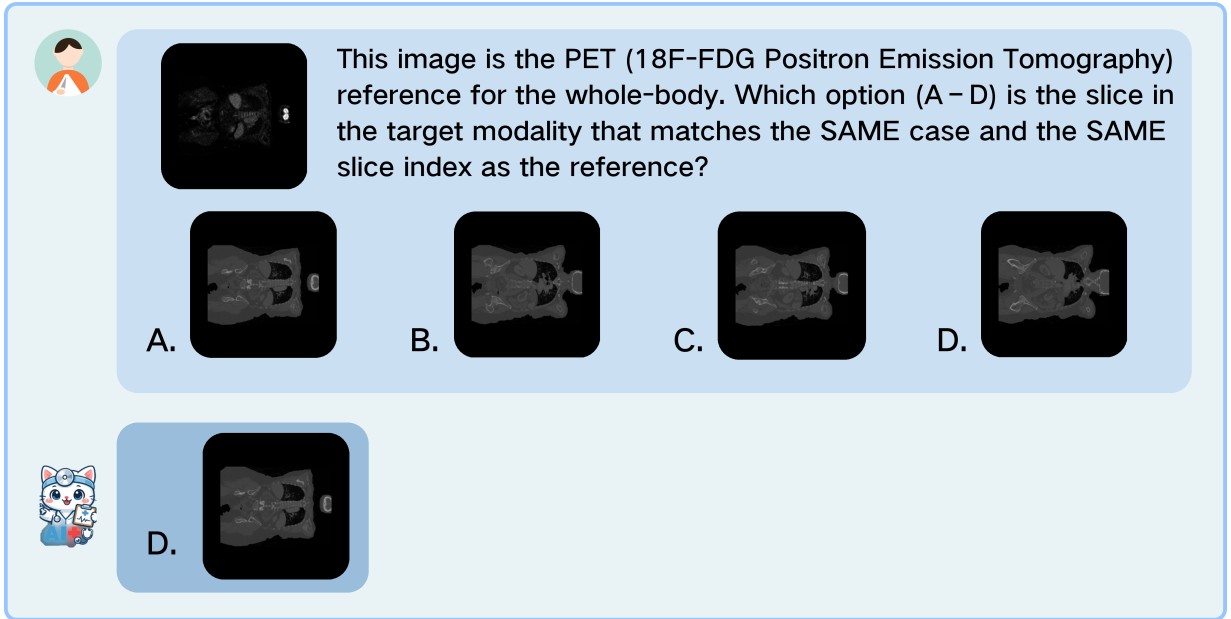

*Figure 18.* Visual question answering example demonstrating cross-modality slice alignment from PET to CT.

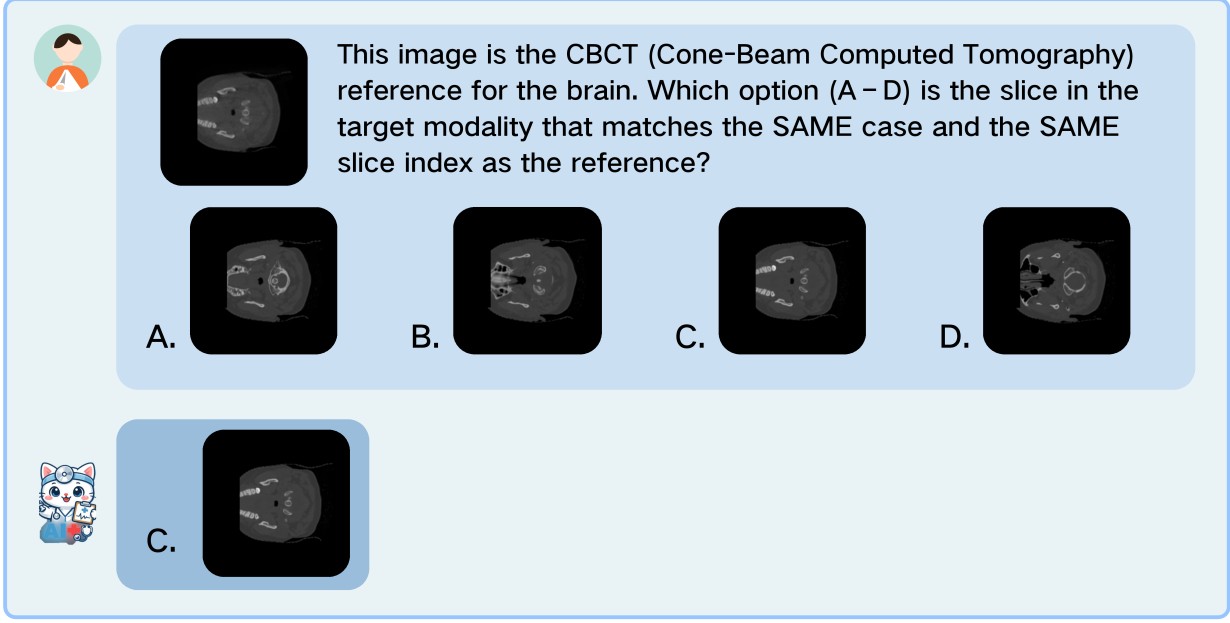

*Figure 19.* Visual question answering example demonstrating cross-modality slice alignment from CBCT to CT.

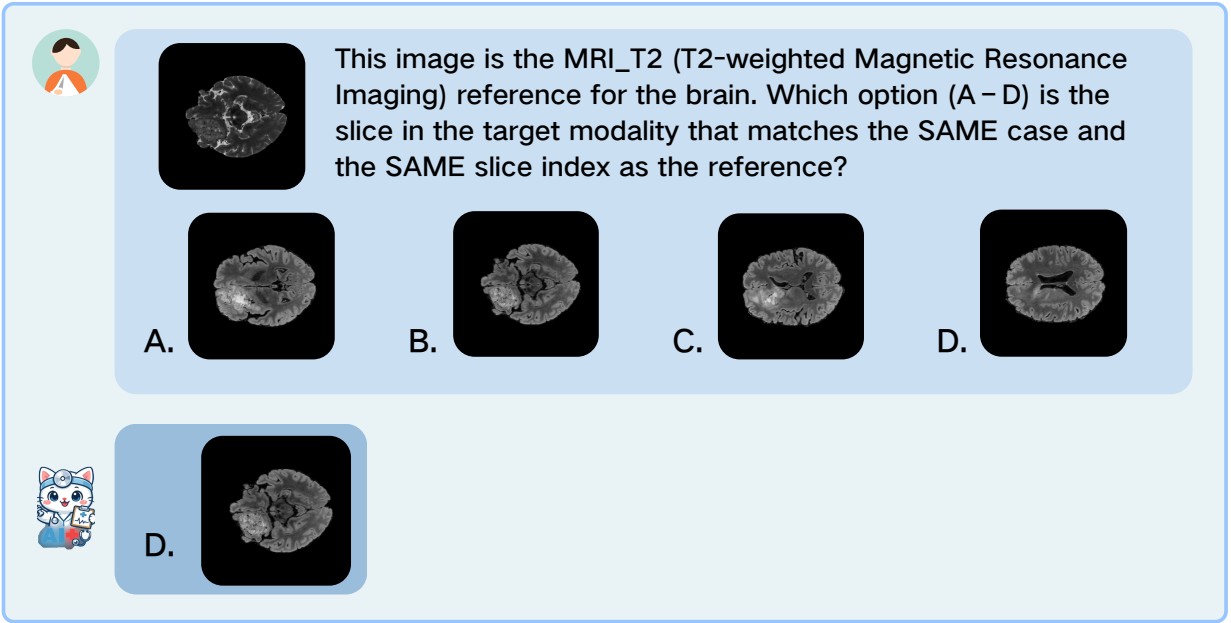

*Figure 20.* Visual question answering example demonstrating cross-modality slice alignment from T2 to FLAIR.

## E.2. Modality Identification

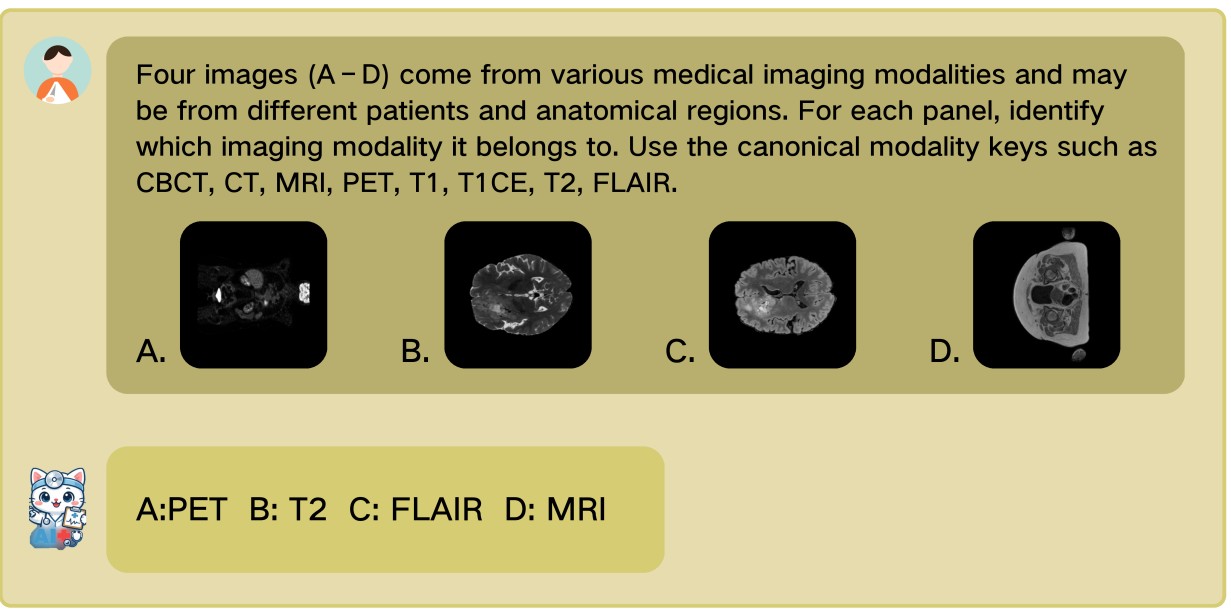

*Figure 21.* Visual question answering example demonstrating the identification of various medical imaging modalities.

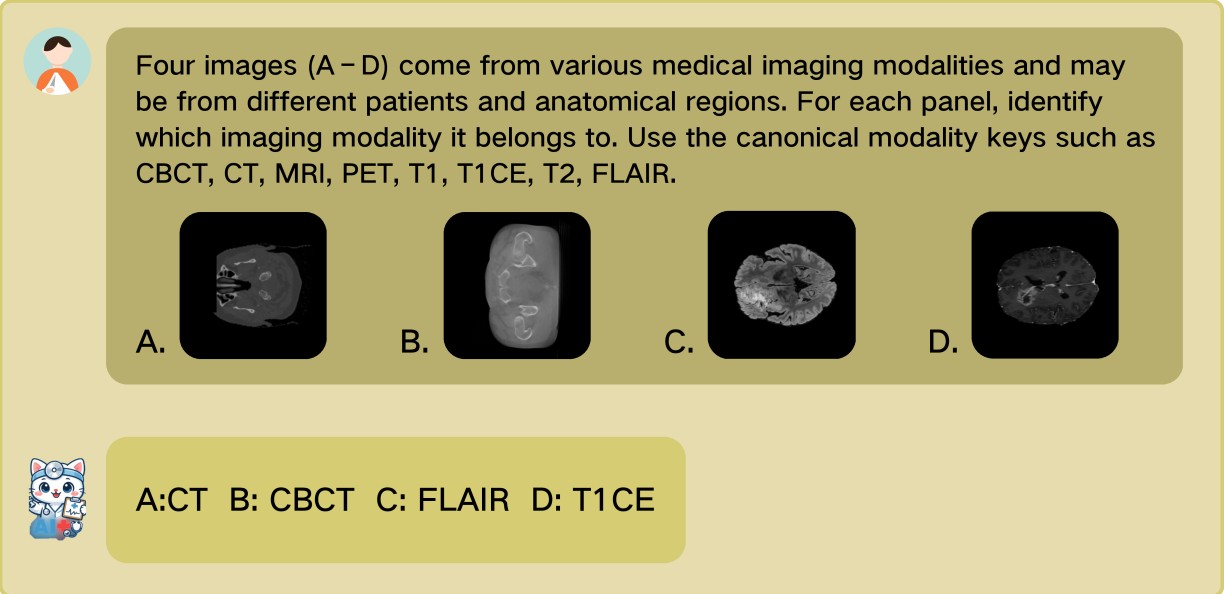

Four images (A – D) come from various medical imaging modalities and may be from different patients and anatomical regions. For each panel, identify which imaging modality it belongs to. Use the canonical modality keys such as CBCT, CT, MRI, PET, T1, T1CE, T2, FLAIR.

A. B. C. D.

A:CT  B: CBCT  C: FLAIR  D: T1CE

*Figure 22.* Visual question answering example demonstrating the identification of various medical imaging modalities.

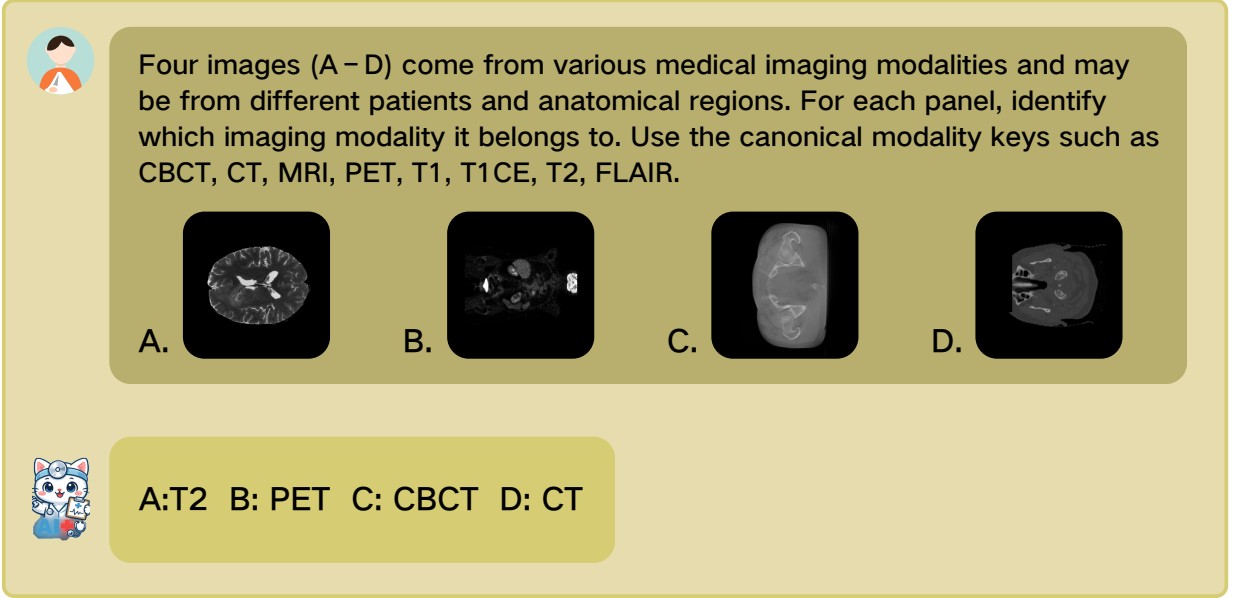

Four images (A – D) come from various medical imaging modalities and may be from different patients and anatomical regions. For each panel, identify which imaging modality it belongs to. Use the canonical modality keys such as CBCT, CT, MRI, PET, T1, T1CE, T2, FLAIR.

A. B. C. D.

A:T2  B: PET  C: CBCT  D: CT

*Figure 23.* Visual question answering example demonstrating the identification of various medical imaging modalities.

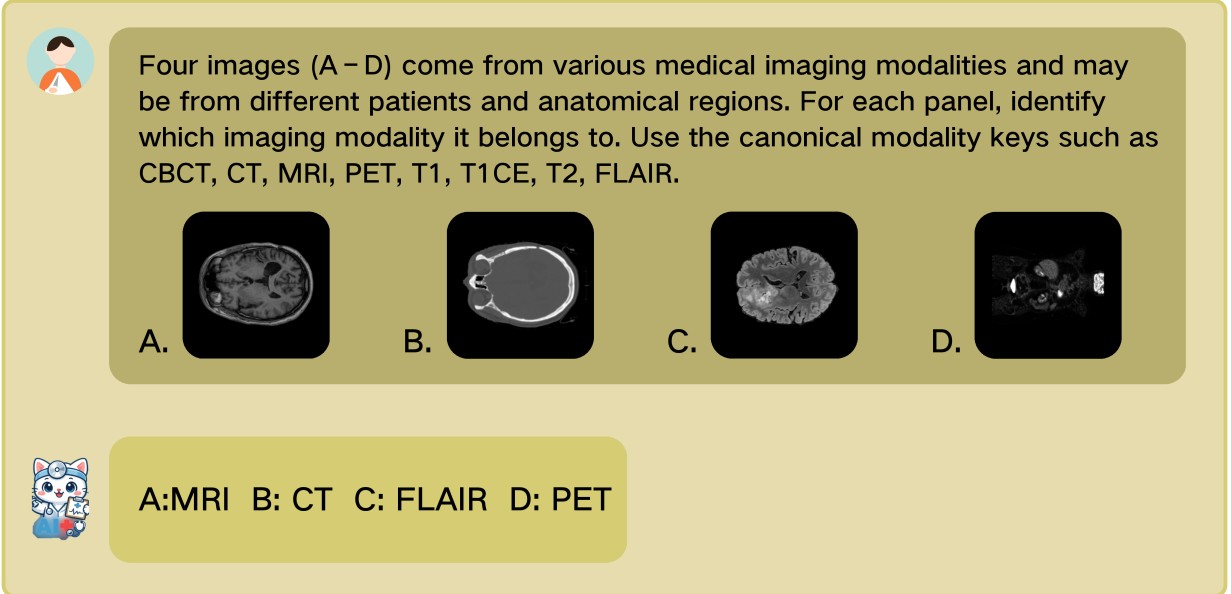

*Figure 24.* Visual question answering example demonstrating the identification of various medical imaging modalities.

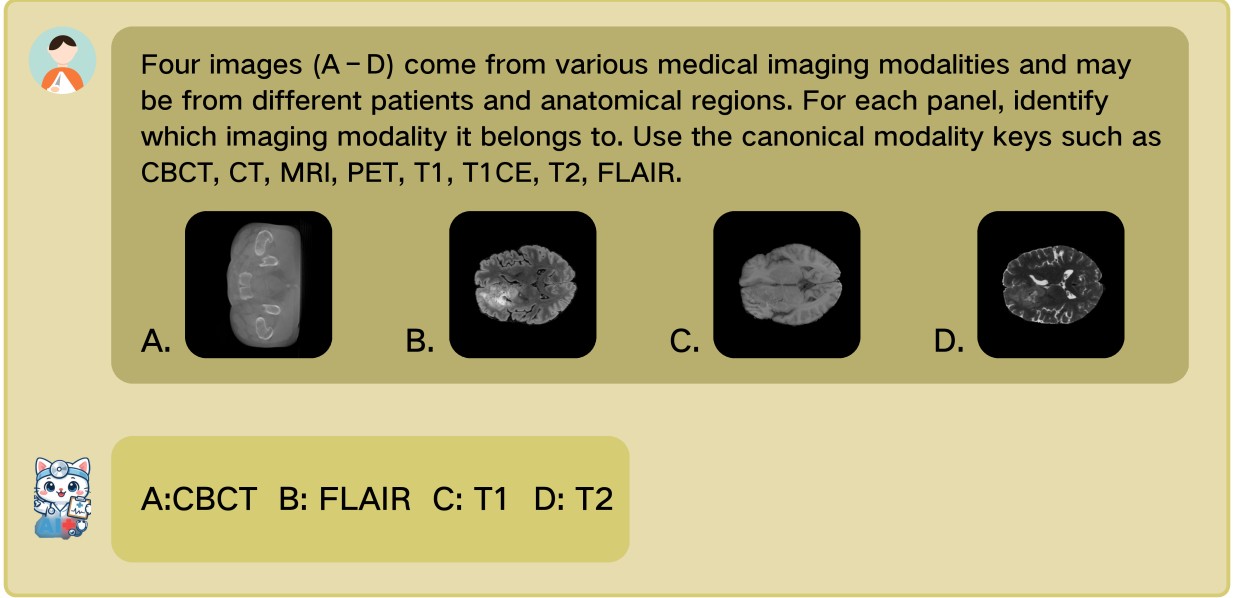

*Figure 25.* Visual question answering example demonstrating the identification of various medical imaging modalities.

## E.3. Transformation Instruction Alignment

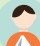 Two images (source and target) are provided in order. Source is the input; target is the aligned counterpart from the SAME case and SAME slice index, but with different contrast characteristics. Read the candidate prompt descriptions and select the one that BEST describes this transformation.

Source. 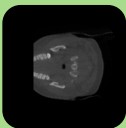      Target. 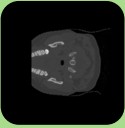

A. Keep anatomy, voxel grid, field-of-view, slice position, and lesion morphology strictly unchanged (1:1 mapping). Modify only photometric/texture properties appearance constraints Lower soft-tissue contrast vs diagnostic ; compression acceptable (air near-black; bone very bright, mild clipping allowed). Add realistic but moderate texture: granular noise; mild cone-beam streaks/flare; slight ring artifacts—must not obscure anatomy. Gentle scatter shading/cupping permissible but avoid vignetting that changes boundaries.• Preserve hardware geometry; avoid saturation/erasure. No hallucinated structures or geometric warping.

B. Translate the input image to output with robust CSF suppression. Maintain voxel-wise alignment and all anatomical boundaries; enforce dark ventricles and sulcal CSF, while periventricular and edematous changes remain conspicuous where appropriate. Avoid haloing, ringing, or geometry changes; output a clean, realistic grayscale fluid-suppressed appearance.

C. Task: Keep anatomy, voxel grid, field-of-view, slice position, and lesion morphology strictly unchanged (1:1 mapping). Modify only photometric properties.Standard constraints:• -like behavior: air very dark (~−1000 ), water mid-gray (~0 ), cortical/trabecular bone very bright.• Suppress -specific artifacts: scatter shading/cupping, ring artifacts, truncation, cone-beam streaks/flare.• Edges like reconstruction kernels (sharper than ) without halo/overshoot.• Preserve metal/hardware geometry; reduce only spurious glare around them.No invented anatomy, no contrast-agent-contrast patterns; realistic diagnostic noise.

D. Convert the input image into a fluid-sensitive output contrast. Replace CSF suppression with bright CSF and appropriate fluid-sensitive tissue relationships, keeping edema/lesions visible yet avoiding any contrast-agent signatures. Anatomy, geometry, and slice framing must be unchanged; deliver a natural grayscale fluid-sensitive texture.

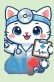 C

*Figure 26.* Visual question answering example identifying the image translation task from CBCT to CT.

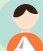

Two images (source and target) are provided in order. Source is the input; target is the aligned counterpart from the SAME case and SAME slice index, but with different contrast characteristics. Read the candidate prompt descriptions and select the one that BEST describes this transformation.

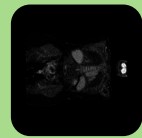

Source.

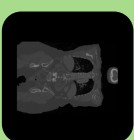

Target.

A: Keep anatomy, voxel grid, field-of-view, slice position, and lesion morphology strictly unchanged (1:1 mapping). Modify ONLY contrast to emulate attenuation. constraints :• Cortical/trabecular bone → bright to very bright; air spaces → near-black.• Soft-tissue ordering: fat < water (mid-gray) < muscle/solid organs < bone.• Remove image blur/halo appearance; produce realistic diagnostic noise/edge sharpness.No contrast-agent-contrast patterns or invented anatomy; preserve exact geometry.

B: Synthesize a contrast-agent output image rendering from the pre-contrast input image. Hold anatomy, pixel grid, and field-of-view fixed; modify only contrast behavior. Introduce plausible contrast-agent enhancement where anatomically expected (e.g., vessels, dura, genuinely enhancing tumor rim/solid components), while non-enhancing tissues remain near baseline soft-tissue baseline and CSF stays dark. Do not hallucinate or resize lesions, and keep noise/texture realistic in grayscale.

C: Keep anatomy, voxel grid, field-of-view, slice position, and lesion morphology strictly unchanged (1:1 mapping). Change only contrast to emulate attenuation. constraints :• Ilium/ischium/pubis/sacrum → cortical/trabecular bone bright to very bright.• Bowel/rectal gas → near-black.• Bladder/urine or other simple fluids → ~water density (mid-gray). Fat hypodense vs water; muscle/solid organs slightly denser than fat.• Stones/calcifications (if implied) → hyperdense; otherwise do NOT invent.No structure addition/removal, no contrast-agent-contrast patterns; realistic texture.

D: Reconstruct the output image appearance from the CSF-suppressed input image. Restore normal baseline soft-tissue contrast (dark CSF; white matter relatively bright vs gray matter) without creating any enhancement. Keep geometry and fine anatomy identical; output should be grayscale with realistic texture.

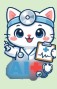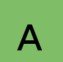

A

*Figure 27.* Visual question answering example identifying the image translation task from PET to CT.

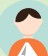 Two images (source and target) are provided in order. Source is the input; target is the aligned counterpart from the SAME case and SAME slice index, but with different contrast characteristics. Read the candidate prompt descriptions and select the one that BEST describes this transformation.

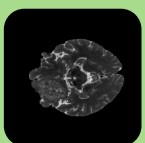

Source.

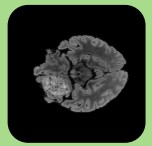

Target.

A: Convert this contrast-agent input image to output image. Remove enhancement cues yet preserve every boundary; enforce CSF nulling so ventricles and sulci are dark while pathologic hyperintensities remain visible where appropriate. No cropping/padding or morphological edits; produce a realistic grayscale fluid-suppressed texture.

B: Produce a output image counterpart from the input image that suppresses free fluid while keeping edema/lesions conspicuous. Do not introduce enhancement-like effects; instead, preserve alignment, resolution, and tissue boundaries exactly. CSF in ventricles and sulci should be near-black; maintain a clean, artifact-free fluid-suppressed look.

C: Keep anatomy, voxel grid, field-of-view, slice position, and lesion morphology strictly unchanged (1:1 mapping). Change only contrast to emulate attenuation. constraints:• Ilium/ischium/pubis/sacrum → cortical/trabecular bone bright to very bright.• Bowel/rectal gas → near-black.• Bladder/urine or other simple fluids → ~water density (mid-gray). Fat hypodense vs water; muscle/solid organs slightly denser than fat.• Stones/calcifications (if implied) → hyperdense; otherwise do NOT invent.No structure addition/removal, no contrast-agent-contrast patterns; realistic texture.

D: Undo contrast-agent enhancement to reconstruct the non-contrast output image counterpart of this contrast-agent input slice. Return vessel/meningeal/enhancing regions toward baseline baseline soft-tissue signal while keeping identical spatial mapping and sharp anatomical edges. No smoothing away detail, no deformation, and CSF should remain dark in the final image.

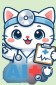 B

*Figure 28.* Visual question answering example identifying the image translation task from T2 to FLAIR.

