# OpenReview forum: "SynerMedGen: Synergizing Medical Multimodal Understanding with Generation via Task Alignment"
_ICML.cc/2026/Conference — ICML 2026 regular_

### Official Review · Reviewer_h2U6 · 2026-03-12

**Soundness:** 2
**Presentation:** 3
**Significance:** 2
**Originality:** 3
**Overall Recommendation:** 4
**Confidence:** 4

**Summary:**

This paper proposes SynerMedGen, a unified framework for medical multimodal understanding and image generation. The key idea is to align understanding supervision with generation objectives by deriving generation‑aligned understanding tasks directly from paired medical image synthesis data. Three understanding tasks—Conditional Target Selection, Modality Identification, and Transformation Instruction Alignment—are designed to encourage slice‑level correspondence, explicit modality awareness, and direction‑aware transformation constraints. A two‑stage training strategy is adopted, where generation‑aligned understanding is first learned and then transferred to latent‑space conditional image synthesis. Extensive experiments across 22 cross‑modality medical image synthesis tasks demonstrate strong zero‑shot performance and consistent improvements over both general unified medical MLLMs and specialized medical image synthesis models.

**Compliance With Llm Reviewing Policy:**

Affirmed.

**Final Justification:**

The rebuttal addressed my concerns regarding clinical relevance and pathology preservation through additional experiments. While 2D slice-wise processing remains a limitation for 3D volumes, the empirical performance is strong. I recommend discussing these 3D limitations in the final manuscript. I have raised my score to Weak Accept.

**Key Questions For Authors:**

1. How does the proposed 2D framework ensure 3D volumetric consistency in clinical scenarios?
2. Can the model preserve fine-grained pathological landmarks (e.g., small tumors or subtle lesions) during synthesis?
3. Does the alignment with generation tasks compromise the model's performance on general medical VQA benchmarks? Could the authors report performance on standard medical VQA benchmarks？

**Limitations:**

The proposed approach focuses on paired 2D medical image translation and does not address 3D volumetric consistency or pathology‑aware generation. The understanding tasks emphasize structural and modality alignment rather than clinical semantics. Additionally, the claimed architecture‑agnostic nature of the method is not empirically demonstrated.

**Strengths And Weaknesses:**

Strengths:
1. The problem is well-described and the motivation is clear. This paper points out a significant shortcoming in the field of unified medical multimodal modeling: existing understanding supervision mechanisms are often mismatched with the needs of medical image synthesis. The proposed generative aligned understanding principle is conceptually clear and reasonable.
2. The understanding tasks are carefully designed. The three tasks proposed in this paper are directly derived from paired synthetic data and aim to address key challenges in medical image translation.
3. The method is evaluated on a wide range of cross-modal synthesis tasks across multiple datasets and anatomical regions.

Weaknesses:
1. While the paper claims unified medical understanding and generation, the generation tasks are primarily limited to paired cross‑modality image translation. The proposed understanding tasks mainly capture structural correspondence and modality‑level constraints, but do not reflect higher‑level clinical or pathological understanding (e.g., lesion semantics, disease progression, or diagnosis).
2. The method operates on 2D slices, whereas many practical challenges in medical image translation arise in 3D volumetric settings, such as maintaining inter‑slice consistency. No analysis or experiments are provided for 3D volumes.
3. Visual comparisons only involve general unified models that are not trained for paired medical image translation, which may be an unfair comparison. Similarly, understanding performance is evaluated only on task‑specific benchmarks derived from the same training paradigm.
4. Although the authors argue that the proposed principle is architecture‑agnostic, all experiments are conducted within a single unified framework, without validation on alternative multimodal architectures.

---

> ### Author Rebuttal · Authors · 2026-03-31
>
> We thank the reviewer for recognizing our clear motivation, conceptually sound method, and comprehensive experiments. Our detailed responses and concise new results are provided below. Full results are available at **[link](https://anonymous.4open.science/r/ICMLR4-E031)**.
> > W1&Q2 Generation scope & understanding level
>
> **Generation scope.** We added **two new generation tasks**: CT denoising on LDCT and MRI super-resolution on IXI. Table R1 shows that SynerMedGen performs best on both tasks, indicating that the generation-aligned understanding principle is not limited to cross-modality translation. Our intention in the paper is to address a broader shortcoming of unified medical multimodal modeling through a general principle, while using cross-modality synthesis as the main testbed because it is both common and challenging. We will revise the wording accordingly and present broader validation as future work.
>
> Table R1: CT denoising & MRI super-resolution SSIM.
> |Method|Denoising|Super-resolution|
> |---|---|---|
> |TA-GAN|78.16|—|
> |VDSR|—|91.42|
> |HealthGPT|67.37|88.66|
> |UniMedVL|68.89|89.07|
> |Bagel|29.31|21.87|
> |SynerMedGen-GAU|58.79|80.77|
> |SynerMedGen|79.78|93.69|
>
> **Understanding level.** We agree that higher-level pathological semantics are important. While our understanding tasks do not explicitly supervise disease progression, they are effective in preserving pathology rather than only low-level appearance.
>
> * **Pathology preservation is encouraged by design.** In CTS, the model must identify the true same-slice target against hard negatives, which requires preserving case-specific anatomical/pathological content. In TIA, the instruction (Appendix E.3) explicitly separates what should remain invariant (e.g., anatomy, lesion morphology) from what should change.
>
> * **Brain tumor segmentation experiment.** Using 100 paired BraTS cases (70%/10%/20% split), we trained and evaluated nnUNet under three input settings in Table R2. Adding synthetic T1CE improves DSC from 56.25 to 62.48, close to 64.83 obtained with real T1CE, indicating that synthesized images retain tumor-relevant cues.
>
> * **Clinical utility for cancer staging.** We incorporated a private head-and-neck MRI T2→T1CE dataset (405 cases, 12,150 slices) for training. Then a radiologist compared synthetic and real T1CE for cancer staging. The staging agreement between synthetic and real T1CE was high (weighted kappa coefficient 0.86/0.82 for T/N category), indicating that SynerMedGen preserves clinically meaningful pathological information.
>
> Table R2: BraTS segmentation.
> |Method|DSC|
> |---|---|
> |T1+T2+FLAIR|56.25|
> |T1+T2+FLAIR+synthetic T1CE|62.48|
> |T1+T2+FLAIR+real T1CE|64.83|
>
> > W2&Q1 3D analysis
>
> * We would like to clarify that, although SynerMedGen is trained on 2D slices, all reported metrics are computed on reconstructed 3D volumes. We perform slice-wise inference, stack the outputs back into full volumes, and **evaluate all methods** at the **3D volume** level. We will clarify this in the revision.
> * SynerMedGen outperforms 3D medical image synthesis methods RCD across all tasks.
> * Our method is not unaware of volumetric structure. In CTS, hard negatives are sampled from neighboring slices within the same volume, forcing the model to distinguish the true paired slice from anatomically similar but non-matching slices, encouraging fine-grained inter-slice correspondence.
> * In the segmentation and clinical study of our response to your W1&Q2 above, the synthesized images stacked as 3D volumes have good utility.
> * Although these results suggest good 3D volumetric consistency, we agree that SynerMedGen is not a full 3D framework, which will be extended as important future work.
>
>
> > W3&Q3 Visual comparison & standard medical VQA benchmarks
>
> **Visual comparison.**  We added more visual comparisons with specialized synthesis methods in the **[link](https://anonymous.4open.science/r/ICMLR4-E031)**.
>
> **Standard medical VQA benchmarks.** We directly tested SynerMedGen on standard medical VQA benchmarks. Table R3 shows that SynerMedGen achieves better performance, indicating that generation-aligned understanding does not weaken general medical VQA ability.
>
> Table R3: Medical VQA.
> |Method|VQA-RAD|SLAKE|OmniMedVQA|MMMU-Med|
> |---|---|---|---|---|
> |InternVL2|49.0|50.1|54.5|43.3|
> |Bagel|60.1|58.9|71.1|46.5|46.5|
> |HealthGPT|58.3|64.5|74.4|49.2|
> |SynerMedGen|61.3|69.4|78.9|50.1|
>
> > W4 Other architecture
>
> We additionally used Janus-Pro as a backbone. In Table R4, the same observations of effectiveness hold. So the benefit of generation-aligned understanding is not confined to Bagel. Our intended claim is that the key idea of deriving understanding supervision directly from the generation data is general. We agree that architecture-agnostic is a strong claim and will tone down this wording.
>
> Table R4. Janus-Pro backbone.
> |Method|SSIM|
> |---|---|
> |Janus-Pro|26.32|
> |SynerMedGen-GAU (Janus-Pro)|67.42|
> |SynerMedGen-UCG (Janus-Pro)|80.60|
> |SynerMedGen (Janus-Pro)|84.94|

---

> > ### Author Rebuttal · Reviewer_h2U6 · 2026-04-04
> >
> > Thank you for your response. My concerns are fully addressed, so I will increase my score.

---

> > > ### Author Response · Authors · 2026-04-06
> > >
> > > Thank you for reviewing our rebuttal and confirming that your concerns have been fully addressed. We greatly appreciate your time and constructive comments.

---

### Official Review · Reviewer_A1VB · 2026-03-13

**Soundness:** 3
**Presentation:** 3
**Significance:** 2
**Originality:** 3
**Overall Recommendation:** 4
**Confidence:** 4

**Summary:**

This paper studies what type of “understanding” benefits medical image generation, especially cross-modality synthesis. The authors propose SynerMedGen, a Bagel-style unified framework that introduces three generation-aligned understanding tasks and transfers the learned representations to conditional generation through a two-stage training strategy. Experiments cover 22 synthesis tasks across multiple datasets. While the idea is reasonable and empirically explored, the evidence mainly relies on pixel-level metrics and relatively close evaluation settings.

**Compliance With Llm Reviewing Policy:**

Affirmed.

**Final Justification:**

After reviewing the rebuttal, I will keep my original rating of Weak Accept.

**Key Questions For Authors:**

1. Do the gains truly arise from understanding–generation synergy, or simply from repackaging large paired synthesis data into auxiliary supervision? If similar stage-I tasks with the same data scale but without explicit generation alignment were used, how much improvement would remain?
2. How can the authors verify that lesions and clinically important structures are better preserved, rather than merely improving intensity similarity reflected in SSIM/PSNR? Stronger lesion-level, organ-level, or downstream diagnostic evaluations would be needed.
3. Since CTS/MI/TIA are derived from paired synthesis data, has the model effectively already learned the transformation rules during stage I? If so, the reported zero-shot improvements may resemble route-aware pretraining rather than true zero-shot generation.
4. Why is the principle only validated on cross-modality synthesis? If generation-aligned understanding is a general unified modeling principle, it should also be tested on other medical generation tasks such as report-grounded generation, image editing, inpainting, or super-resolution.
5. Will the method remain effective when paired data are limited or imperfectly aligned? The current design relies heavily on slice correspondence and transformation routes, which may not hold in more realistic weakly paired clinical settings.
6. Is a unified model necessarily preferable to a modular pipeline combining a strong specialized synthesis model with a lightweight understanding module? The paper shows performance gains but does not sufficiently discuss the added complexity, training cost, or potential failure modes of the unified design.

**Limitations:**

Yes. The paper discusses limitations and potential risks, though the discussion could be strengthened by further addressing clinical evaluation, data dependence, and generalization to weaker or imperfectly paired medical datasets.

**Strengths And Weaknesses:**

Strengths:
1. The problem formulation is clear and well motivated. The paper identifies the mismatch between understanding supervision and synthesis objectives in unified medical MLLMs and decomposes it into slice correspondence, modality controllability, and transformation direction.
2. The method design is simple and naturally aligned with the task. The proposed understanding tasks are derived directly from paired synthesis data and closely match the information required for cross-modality synthesis.
3. The empirical study is broad and includes controlled validation. Experiments across 22 tasks and comparisons with fixed architecture and training support that the gains mainly come from task alignment rather than confounding factors.

 Weaknesses:
1. The novelty mainly lies in task design rather than model mechanism. The method largely builds on an existing unified architecture and improves performance through redesigned understanding tasks and a two-stage training scheme, which is closer to task/interface engineering than a fundamentally new unified modeling mechanism.
2. Evidence for medical validity is limited, with evaluation relying heavily on SSIM, PSNR, and MAE. These metrics alone cannot establish lesion fidelity, structural consistency, or clinical usefulness, and stronger clinical evaluations are missing.
3. The fairness of comparisons with unified baselines remains uncertain. HealthGPT and UniMedVL are evaluated using public checkpoints, but differences in input format, task support, prompting, and modality coverage may prevent optimal adaptation to the current setup.

---

> ### Author Rebuttal · Authors · 2026-03-31
>
> We appreciate the reviewer's positive feedback on our well-motivated problem, task-aligned method and broad empirical study. Our responses with concise new tables are below. Full results are at **[link](https://anonymous.4open.science/r/ICMLR3-3461)**.
>
> > W1 Contribution
>
> Although we do not introduce a new modeling mechanism, our contributions lie in two important aspects:
> * Identify understanding–generation misalignment as a key bottleneck in unified medical models.
> * Propose generation-aligned understanding, where CTS/MI/TIA provides structured supervision to align understanding learning with the generation objective.
>
> > W2&Q2 Lesions & clinical validity
>
> We added two lesion-related evaluations:
> * **Brain tumor segmentation.** Using 100 paired BraTS cases (70/10/20 split), we trained and evaluated nnUNet under three input settings in Table R1. Synthetic T1CE improves DSC from 56.25 to 62.48, close to 64.83 with real T1CE, indicating preservation of tumor-relevant cues.
> * **Clinical utility for cancer staging.** We incorporated a private head-and-neck MRI T2→T1CE dataset (405 cases, 12,150 slices) for training. Then a radiologist compared synthetic and real T1CE for cancer staging. The staging agreement between synthetic and real T1CE was high (weighted kappa coefficient 0.86/0.82 for T/N category), preserving clinically meaningful pathology.
>
> Table R1: BraTS segmentation.
> |Method|DSC|
> |---|---|
> |T1+T2+FLAIR|56.25|
> |T1+T2+FLAIR+synthetic T1CE|62.48|
> |T1+T2+FLAIR+real T1CE|64.83|
>
> > W3 Baseline fairness
>
> Since training code for HealthGPT and UniMedVL is unavailable, we used their public checkpoints, but clarify the following fair comparisons:
> * Table 1-2 of the paper shows that our gains hold on datasets overlapping with their training data (Brain/Pelvis MRI-CT for HealthGPT; BraTS for UniMedVL).
> * In Sec. 4.3, we keep the same architecture, data, and training, and just replace our understanding tasks with HealthGPT-style tasks. Fig. 4 shows that our design yields clear gains.
> * For generalization test in Fig. 8, all methods use SynthRAD2023 during training and none see SynthRAD2025. Our method still performs best.
>
> > Q1 Source of gains
>
> We scaled HealthGPT understanding data from 638K to 1M (adding PMC-VQA, PMC-Inline, and Medical-diff-VQA). The gain is minimal in Table R2&R3, far below our method. Thus, the gain comes mainly from generation alignment, not supervision scale.
>
> Table R2: Stage Ⅰ.
> |Method|SSIM|
> |---|---|
> |Bagel|29.14|
> |Traditional understanding (638K)|39.37|
> |Traditional understanding (1M)|39.95|
> |SynerMedGen-GAU|74.91|
>
> Table R3: Stage Ⅰ+Ⅱ.
> |Method|SSIM|
> |---|---|
> |Traditional understanding (638K)|83.56|
> |Traditional understanding (1M)|83.69|
> |SynerMedGen|86.59|
>
> > Q3 Zero-shot meaning
>
> * Our intended claim is that Stage I provides generation-aware but non-generative supervision, which improves downstream synthesis even before Stage II training. We will clarify this scope.
> * The Stage I gain is not mere route memorization. In Fig. 7, SynerMedGen-GAU performs strongly on unseen MyoPS, suggesting transferable generation-beneficial priors.
> * We further tested an unseen private head-and-neck dataset, where SynerMedGen-GAU also performs best in Table R4.
>
> Table R4. Head-and-neck dataset.
> |Method|SSIM|
> |---|---|
> |Bagel|42.97|
> |HealthGPT|55.29|
> |UniMedVL|49.14|
> |SynerMedGen-GAU|70.38|
>
> > Q4 Generation scope
>
> We added CT denoising on LDCT and MRI super-resolution on IXI. Table R5 shows that SynerMedGen performs best, indicating that the generation-aligned understanding principle is not limited to cross-modality translation.
>
> Table R5: CT denoising & MRI super-resolution SSIM.
> |Method|Denoising|Super-resolution|
> |---|---|---|
> |TA-GAN|78.16|—|
> |VDSR|—|91.42|
> |HealthGPT|67.37|88.66|
> |UniMedVL|68.89|89.07|
> |Bagel|29.31|21.87|
> |SynerMedGen-GAU|58.79|80.77|
> |SynerMedGen|79.78|93.69|
>
> > Q5 Limited/weak pairing
>
> We tested two harder settings (Table R6).
> * Limited data: using only 10% data, SynerMedGen still outperforms baselines.
> * Weak pairing: with only 5% correctly paired and 5% weakly paired slices, SynerMedGen still improves over UCG (55.36 vs. 51.02).
> Thus, the generation-aligned understanding remains effective under both substantial data reduction and imperfect alignment.
>
> Table R6. Limited/weak pairing.
> |Method|SSIM|
> |---|---|
> |Bagel|29.14|
> |SynerMedGen-GAU（10%)|45.81|
> |SynerMedGen-UCG (10%)|57.57|
> |SynerMedGen (10%)|61.74|
> |SynerMedGen-UCG (5% weakly paired)|51.02|
> |SynerMedGen (5% weakly paired)|55.36|
>
> > Q6 Modular pipeline
>
> We trained a modular baseline using ControlNet-style Stable Diffusion with CLIP-based conditioning. It performs much worse than SynerMedGen (Table R7), suggesting that a lightweight modular interface cannot reliably convey slice correspondence, modality control, and pathology preservation, which SynerMedGen learns through shared generation-aligned representations.
>
> Table R7. CLIP-based modular pipeline.
> |Method|SSIM|
> |---|---|
> |CLIP-based|43.56|
> |SynerMedGen|86.59|

---

> > ### Author Rebuttal · Reviewer_A1VB · 2026-04-03
> >
> > Thank you for your thorough rebuttal. My concerns have been addressed, and I will keep my rating as weak accept.

---

> > > ### Author Response · Authors · 2026-04-06
> > >
> > > We are grateful for your positive feedback and glad to hear that our response adequately addressed your concerns. Thank you for your time and thoughtful evaluation.

---

### Official Review · Reviewer_a3oa · 2026-03-13

**Soundness:** 3
**Presentation:** 4
**Significance:** 3
**Originality:** 3
**Overall Recommendation:** 5
**Confidence:** 3

**Summary:**

SynerMedGen proposes a unified framework that aligns medical image understanding tasks with generation tasks. By designing understanding objectives that directly support image synthesis, the model learns representations that benefit both analysis and generation. A two-stage training strategy and the new SynerMed dataset demonstrate improved performance on multiple medical image generation tasks.

**Compliance With Llm Reviewing Policy:**

Affirmed.

**Final Justification:**

After reading the rebuttal from the authors, my questions have been addressed.

In general I think the proposed approach is interesting and general. Looking forward to seeing more results applied on broader multimodal fields.

**Key Questions For Authors:**

1. Dataset construction and preprocessing details
The paper introduces the SynerMed dataset and performs experiments across multiple medical image synthesis tasks, but the details of the data preparation process are not fully clear. Specifically, how were the multimodal pairs (e.g., CT–MRI, PET–CT) curated and aligned across datasets, and what preprocessing steps (e.g., registration, normalization, resolution standardization) were applied before training? Clarifying this would help assess whether the reported improvements stem from the proposed method or from dataset-specific preprocessing. A detailed explanation could strengthen the reproducibility and credibility of the experimental results.

2. Fairness and robustness of experimental comparisons
The experimental results show consistent improvements over several baselines, but it is not entirely clear whether all methods were trained under fully comparable settings (e.g., same datasets, data splits, training iterations, and model capacity). Could the authors clarify whether all baselines were retrained under the same experimental protocol as SynerMedGen, or whether some results were taken from prior papers? This clarification would help determine the fairness of the comparisons and could impact the confidence in the reported performance gains.

**Limitations:**

The current experiments focus on a limited set of medical imaging modalities and generation tasks, leaving the effectiveness of the approach on a broader range of modalities and tasks unclear.

**Strengths And Weaknesses:**

Strengths
1. Clear and well-motivated problem formulation
The paper identifies an important limitation in existing medical multimodal models—understanding and generation are typically optimized separately, which prevents meaningful synergy between the two tasks.

2. Novel task-alignment perspective
The proposed idea of generation-aligned understanding is conceptually interesting and provides a principled way to design understanding objectives that directly benefit generative tasks.

3. Unified framework with practical training strategy
SynerMedGen presents a coherent framework with a two-stage training pipeline that is easy to integrate with existing multimodal and generative models.

4. Strong empirical evaluation
The method is evaluated on a large set of medical image synthesis tasks and datasets, demonstrating consistent improvements over strong baselines.

Weaknesses
1. Ablation studies could be more thorough
It would be helpful to isolate the contribution of each proposed generation-aligned understanding task to better understand which components drive the improvements.

2. Generalization beyond medical imaging remains unclear
The framework is demonstrated only in the medical domain, and it is unclear how well the task-alignment principle would transfer to other multimodal generation settings.

---

> ### Author Rebuttal · Authors · 2026-03-31
>
> We thank the reviewer for the positive feedback regarding the well-motivated problem, novel task-alignment perspective, coherent framework, and strong empirical evaluation of our work. We address the comments point by point below.
>
> > W1 Task ablation
>
> We thank the reviewer for the important suggestion. In addition to the progressive ablation included in the paper (CTS → CTS+MI → CTS+MI+TIA), we now added all the individual and paired task ablations, evaluated after both Stage I and Stage I+II training. Results on Tables R1&R2 show that each task contributes independently, and the full combination obtains the greatest improvement. Full results are available at **[link](https://anonymous.4open.science/r/ICMLR2-98D5)**.
>
> Table R1. Stage Ⅰ.
> |Method|SSIM|
> |---|---|
> |Bagel|29.14|
> |CTS|61.83|
> |MI|50.72|
> |TIA|61.86|
> |CTS+MI|68.92|
> |CTS+TIA|69.00|
> |MI+TIA|66.75|
> |CTS+MI+TIA (Ours)|74.91|
>
> Table R2. Stage Ⅰ+Ⅱ.
> |Method|SSIM|
> |---|---|
> |SynerMedGen-UCG|82.62|
> |CTS|83.79|
> |MI|83.34|
> |TIA|84.40|
> |CTS+MI|85.16|
> |CTS+TIA|85.77|
> |MI+TIA|85.22|
> |CTS+MI+TIA (Ours)|86.59|
>
> > W2 Beyond medical generalization
>
> How to make understanding and generation mutually beneficial remains an important open question in unified multimodal models. We use medical imaging as a particularly meaningful testbed because understanding–generation misalignment in the medical unified model is significant. In medical image synthesis, the model must preserve clinically important anatomy and pathology while changing only modality-dependent appearance, making the alignment problem important.
>
> Conceptually, aligning understanding tasks with generation objectives is not restricted to medical data. However, whether this principle transfers equally well to other multimodal generation settings remains a valuable empirical question to study. We have not included such experiments given the limited rebuttal time. We will therefore keep our claim appropriately scoped to the medical domain and highlight broader validation as an important direction for future work.
>
> > Q1 Dataset preprocessing details
>
> The SynerMed dataset is built by integrating multiple public medical image synthesis datasets consisting of paired data and organizing them into 22 synthesis tasks across CBCT, CT, PET, and multi-sequence MRI. For each source dataset, we form multimodal pairs from the same patient. Before training, we apply a unified preprocessing pipeline to all datasets:
> 1) convert all source volumes into a consistent orientation and format;
> 2) use the native registration/alignment provided by each dataset;
> 3) extract 2D paired slices from aligned multimodal volumes;
> 4) apply modality-specific intensity normalization;
> 5) remove invalid or empty slices;
> 6) construct train/validation/test splits at the patient level to avoid leakage across slices from the same subject.
>
> To further strengthen reproducibility, we will **release our constructed data, the data construction pipeline, and training code** upon publication.
>
> > Q2 Fairness of comparisons
>
> We clarify the experimental setup for fair comparison.
>
> * For **specialized synthesis methods** (e.g., Pix2Pix, SynDiff, and RCD), we retrained each model using the publicly released code, with each subtask trained separately under the same data splits and protocol as SynerMedGen.
>
> * For **unified baselines** HealthGPT and UniMedVL, since their training code is not released, we used their public checkpoints on the same test sets as in our paper. Our gains hold both on datasets overlapping with their training data (Brain/Pelvis MRI-CT for HealthGPT; BraTS for UniMedVL) and on other datasets.
>
> * We also include a **controlled comparison** that does not rely on public checkpoints of HealthGPT and UniMedVL. In Section 4.3, we keep the same architecture, data, and training, and replace only our Stage I generation-aligned understanding tasks with traditional HealthGPT-style understanding tasks. Fig. 4 shows that our design yields clear gains under this controlled setting, directly supporting the effectiveness of task alignment.
>
> > Limitations: more generation tasks
>
> We added two new generation tasks: CT denoising on LDCT and MRI super-resolution on IXI. Table R3 shows that SynerMedGen performs best on both tasks, indicating that the generation-aligned understanding principle is not limited to cross-modality translation. Our intention in the paper is to address a broader shortcoming of unified medical multimodal modeling through a general principle, while using cross-modality synthesis as the main testbed because it is both common and challenging. We will present broader validation as future work.
>
> Table R3: CT denoising & MRI super-resolution SSIM. Full results are available at **[link](https://anonymous.4open.science/r/ICMLR2-98D5)**.
> |Method|Denoising|Super-resolution|
> |---|---:|---:|
> |TA-GAN|78.16|—|
> |VDSR|—|91.42|
> |HealthGPT|67.37|88.66|
> |UniMedVL|68.89|89.07|
> |Bagel|29.31|21.87|
> |SynerMedGen-GAU|58.79|80.77|
> |SynerMedGen|79.78|93.69|

---

> > ### Author Rebuttal · Reviewer_a3oa · 2026-04-04
> >
> > The rebuttal experiments are sound and provide enough evidence for the method.

---

> > > ### Author Response · Authors · 2026-04-06
> > >
> > > Thank you for your thoughtful follow-up and positive assessment. We are pleased that our response has adequately addressed your concerns. We sincerely appreciate your time and constructive feedback.

---

### Official Review · Reviewer_7Pa8 · 2026-03-14

**Soundness:** 3
**Presentation:** 2
**Significance:** 3
**Originality:** 3
**Overall Recommendation:** 4
**Confidence:** 3

**Summary:**

This paper introduces SynerMedGen, a unified framework designed to jointly handle medical multimodal understanding and generation. The authors point out that most existing medical multimodal models treat understanding and generation as separate objectives, which often leads to representations that work well for analysis but contribute little to tasks like image generation or cross-modal synthesis. To address this issue, the paper proposes a principle called Generation-Aligned Understanding, where the understanding capability is directly trained using paired generation data. Based on this idea, the authors design three understanding tasks: CTS, MI, and TIA. The model is trained with a two-stage strategy. First, it performs alignment training for the understanding tasks. Then, it learns conditional generation using a flow matching approach. Experiments show that the model achieves SOTA performance on 22 medical image synthesis tasks, demonstrating strong zero-shot ability and good cross-dataset generalization.

**Compliance With Llm Reviewing Policy:**

Affirmed.

**Key Questions For Authors:**

See Weakness

**Limitations:**

See Weakness

**Strengths And Weaknesses:**

# Strengths

- It explicitly points out that the global semantic understanding characteristic of traditional VQA approaches is insufficient to meet the stringent requirements of medical image synthesis, specifically, dense pixel-level alignment and anatomical structure preservation. This profound insight offers a novel perspective for the design of unified large-scale models.

- The experiments are remarkably comprehensive. In particular, the results of the ablation study, demonstrating that significant improvements in generation tasks can be achieved solely through Stage I comprehension training, provide extremely compelling evidence for the core hypothesis.

# Weaknesses

1. Lack of assessment of the effectiveness of traditional understanding tasks. The paper does not report the model's performance on traditional medical understanding tasks (like disease diagnosis VQA, radiology report generation, lesion grounding).

2. Task 3 (TIA) utilizes textual descriptions to anchor the direction of translation. However, the paper remains vague regarding the specific composition of these "descriptions." Do these instructions consist of simple, fixed templates (like Translate from CT to MRI), or do they incorporate rich semantic information involving pathological and anatomical reasoning?

3. Both the paper's title and introduction emphasize Generation, yet the experimental section consists entirely 100% of cross-modal image translation. I suggest slightly narrowing the scope within the Limitations section or the introduction.

4. How does the model leverage the representations learned in Stage I during Stage II? In Equation (6) of Section 3.3, reference is made solely to relying on source evidence $c$. Specifically, how is the hidden-layer space from the Stage I MoT actually passed to the Stage II Flow Matching prediction network? Is this achieved via cross-attention, or simply through concatenation?

---

> ### Author Rebuttal · Authors · 2026-03-31
>
> We appreciate the reviewer’s positive feedback on our profound insight and novel perspective for the design of unified models and remarkably comprehensive experiments for compelling evidence. We respond to each of the comments in detail below.
>
> > W1: Standard medical VQA benchmarks
>
> We thank the reviewer for this important suggestion. We have added an evaluation of SynerMedGen on four standard medical VQA benchmarks. The results in Table R1 show that SynerMedGen outperforms Bagel and other methods, indicating that generation-aligned understanding tasks do not compromise general medical VQA performance. This might be because our understanding tasks improve fine-grained visual representation learning that is also useful for general medical VQA.
>
> Table R1: Standard Medical VQA benchmarks. Full results are available at **[link](https://anonymous.4open.science/r/ICMLR1-24C7)**.
> |Method|VQA-RAD|SLAKE|OmniMedVQA|MMMU-Med|
> |---|---|---|---|---|
> |InternVL2|49.0|50.1|54.5|43.3|
> |Bagel|60.1|58.9|71.1|46.5|46.5|
> |HealthGPT|58.3|64.5|74.4|49.2|
> |SynerMedGen|61.3|69.4|78.9|50.1|
>
> > W2: TIA descriptions
>
> TIA does not rely on bare templates like “translate CT to MRI” only. Instead, it uses route-level semantically enriched descriptions that encode both invariances (e.g., anatomy, voxel grid, field-of-view, slice position, lesion morphology) and desired changes (e.g., contrast, modality-specific appearance, texture, artifact suppression). The **appendix E.3 examples** in the paper reflect this design, including MRI-specific contrast relationships and CBCT-specific artifact suppression constraints. These descriptions were constructed in a semi-automatic but human-curated way. We used representative paired examples for each route, asked an LLM to draft candidate descriptions, and then manually checked/refined them, including a brief review by clinicians.
>
> > W3: Generation scope
>
> We thank the reviewer for this valuable suggestion. To broaden validation beyond cross-modality translation, we added **two new generation tasks**: CT denoising on LDCT and MRI super-resolution on IXI. Table R2 shows that our SynerMedGen achieves better performance on both tasks, indicating that the proposed generation-aligned understanding principle is not limited to cross-modality translation. Although the current paper still does not cover all medical generation settings. Our intention in the paper is to address a broader shortcoming of unified medical multimodal modeling through a general principle, while using cross-modality synthesis as the main testbed because it is both common and challenging. We will point out this limitation and present broader validation as future work.
>
> Table R2: CT denoising & MRI super-resolution SSIM. Full results are available at **[link](https://anonymous.4open.science/r/ICMLR1-24C7)**.
> |Method|Denoising|Super-resolution|
> |---|---:|---:|
> |TA-GAN|78.16|—|
> |VDSR|—|91.42|
> |HealthGPT|67.37|88.66|
> |UniMedVL|68.89|89.07|
> |Bagel|29.31|21.87|
> |SynerMedGen-GAU|58.79|80.77|
> |SynerMedGen|79.78|93.69|
>
> > W4: Stage-I to Stage-II transfer
>
> In the unified Mixture-of-Transformer-experts (MoT) architecture Bagel, the understanding-oriented text and image tokens and the generation-oriented VAE latents are first projected into a shared MoT hidden space, and understanding and generation interact through the unified token-processing mechanism.
>
> Concretely, in Stage I, CTS/MI/TIA uses the understanding pathway to encourage the model to learn source-conditioned representations that preserve the information required for downstream synthesis. In Stage II, we initialize the unified model from the Stage-I checkpoint. The generation expert then conditions on these source-side representations through attention in the shared MoT space, while predicting the velocity field for the noisy target latent.
>
> So this transfer is not implemented through a separate adapter, nor by simply concatenating a final hidden state. Under the Bagel architecture, text tokens, source-image ViT tokens, and generation-side VAE tokens are mapped into the same MoT hidden space and interact through shared multimodal self-attention with generalized causal masking. As a result, the generation-side tokens used by the Flow Matching network can attend to the preceding source-side text/ViT tokens. Therefore, although Stage I does not directly train the generation objective, it improves the conditioning representation that Stage II relies on for generation.

---

> > ### Author Rebuttal · Reviewer_7Pa8 · 2026-04-08
> >
> > Thanks for the detailed rebuttal, which has addressed my concerns.

---

> > > ### Author Response · Authors · 2026-04-08
> > >
> > > Thank you for your encouraging comments. We are pleased that our rebuttal has addressed your concerns and greatly appreciate your positive feedback.

---

### Decision · Program_Chairs · 2026-04-30

**Decision:**

Accept (regular)

**Comment:**

The paper received all positive recommendations from four reviewers, with three weak accepts and one accept. The contributions proposed in the paper are clear and significant. The unified framework design presents important insights to the community. The experimental verifications are conducted remarkably comprehensively, as mentioned by the reviewers. Based on the consistently positive ratings and the encouraging comments, AC decided to accept this submission.